# Partial thermalisation of a two-state system coupled to a finite quantum bath

**Philip J. D. Crowley[1][*] and Anushya Chandran[2]**

**1** Department of Physics, Massachusetts Institute of Technology,
Cambridge, Massachusetts 02139, USA
**2** Department of Physics, Boston University, Boston, Massachusetts 02215, USA

* pjdc@mit.edu

## Abstract

The eigenstate thermalisation hypothesis (ETH) is a statistical characterisation of eigen-energies, eigenstates and matrix elements of local operators in thermalising quantum systems. We develop an ETH-like ansatz of a partially thermalising system composed of a spin-$\frac{1}{2}$ coupled to a finite quantum bath. The spin-bath coupling is sufficiently weak that ETH does not apply, but sufficiently strong that perturbation theory fails. We calculate (i) the distribution of fidelity susceptibilities, which takes a broadly distributed form, (ii) the distribution of spin eigenstate entropies, which takes a bi-modal form, (iii) infinite time memory of spin observables, (iv) the distribution of matrix elements of local operators on the bath, which is non-Gaussian, and (v) the intermediate entropic enhancement of the bath, which interpolates smoothly between $S = 0$ and the ETH value of $S = \log 2$. The enhancement is a consequence of rare many-body resonances, and is asymptotically larger than the typical eigenstate entanglement entropy. We verify these results numerically and discuss their connections to the many-body localisation transition.



# 1    Introduction

The dynamics of a two-level quantum system coupled to a mesoscale thermal bath is a canonical problem in physics [1–6]. Examples include solid-state qubits coupled to nuclear spins [7–10], trapped ions coupled to phonon modes [11–13], superconducting qubits coupled to magnetic defects [14–20], and many-body localised cold atoms coupled to ergodic inclusions [21, 22].

For infinite temperature random matrix baths, the relevant dimensionless parameter is the *reduced coupling g*, [23–25]

$$g := \frac{J\rho_0}{\sqrt{d}} \qquad \text{(random matrix bath)}. \tag{1a}$$

Above $J$ is the coupling strength between the two-level system (henceforth spin-$\frac{1}{2}$) and the bath, and $\rho_0$ and $d$ are respectively the density of states at maximum entropy and the Hilbert space dimension of the bath. The reduced coupling sets the scale of the first-order (in $J$) correction to an eigenstate, and is given by the ratio of a typical off-diagonal matrix element $J/\sqrt{d}$ to the typical many-body energy level spacing in the bath $1/\rho_0$. For a bath that satisfies the eigenstate thermalisation hypothesis (ETH), the same ratio is given by

$$g := J\sqrt{\tilde{\nu}(h_S)\rho_0} \qquad \text{(ETH bath)}. \tag{1b}$$

Here $\tilde{\nu}(\omega)$ is the spectral function of the coupling operator on the bath, and $h_S$ is the energy splitting of the spin at $J = 0$.

The *strong coupling regime* ($g \gtrsim 1$) is well-studied; here the combined system of the spin and the bath is expected to obey ETH [26–34][1]. At late times, the spin reaches thermal equilibrium. At the opposite extreme, in the *weak coupling regime* ($g \ll 1/d$), the eigenstates of the combined system are described by product states between the spin and bath up to perturbative corrections, and the spin behaves as an isolated system that does not thermalise.

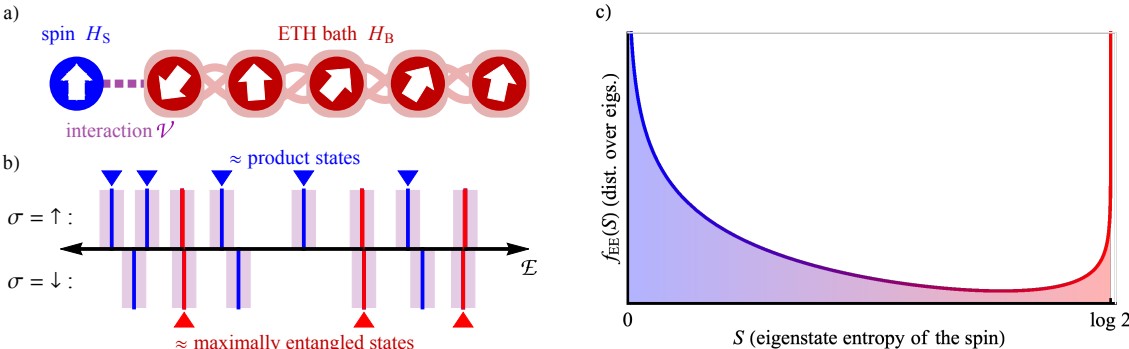

Figure 1: a) *Model*: a spin-$\frac{1}{2}$ intermediately coupled to a many-body quantum bath. b) *A window of the spectrum*: Energy levels in the two spin sectors $\sigma = \uparrow / \downarrow$ of the decoupled Hamiltonian are denoted above/below the energy axis. Two levels strongly hybridise if their energy separation is much smaller than the typical matrix element connecting them (purple collar). Typical levels (blue) do not hybridise, while rare pairs strongly hybridise and form cat states (red). c) *Distribution of spin eigenstate entanglement entropies $f_{EE}$*: $f_{EE}$ is bi-modal with a mode at $S = 0$ ($S = \log 2$) due to the blue (red) states in (b).

---

[1]Formally holding $g$ finite while taking $d \to \infty$ recovers diagonal ETH and not off-diagonal ETH for spin observables. See Sec. 4.1.3.

We develop a statistical theory of spin observables in both eigenstates and dynamical experiments *the intermediate coupling regime* $1/d \lesssim g \ll 1$. Although the majority of eigenstates are nearly product states (blue in Fig. 1b), eigenstate averaged properties are determined by the minority of states involved in rare *many-body resonances* (red). These resonant states are approximately cat states with spin entanglement entropy $S$ close to $\log 2$. The nearly product and cat eigenstates determine two modes in the distribution of $S$ across eigenstates (Fig. 1c, Sec. 4.1). In contrast, in an ETH system, the distribution has a single mode at $S = \log 2$. The spin-bath system thus does not satisfy ETH in the intermediate coupling regime. It is however *partially thermalising*, as spin observables only retain partial memory of initial conditions at late times (Sec. 5).

The spin-bath system functions as a bath with a non-ETH (i.e. non-Gaussian) distribution of off-diagonal matrix elements (Sec. 6) and an enhanced entropy as compared to the bare bath (Sec. 7). The entropy of the spin-bath system probed by a second spin (Fig. 2a) smoothly increases from $\mathcal{S} = \log \rho_0$ in the weak coupling regime, to $\mathcal{S} = \log(2\rho_0)$ in the strong coupling regime. We calculate the entropic enhancement $\Delta \mathcal{S}$ exactly throughout the intermediate regime

$$\Delta \mathcal{S}(J) = 2\log\left(\frac{[|V'_{\alpha\beta}|]}{[|V'_{\alpha\beta}|]_{J=0}}\right), \tag{2}$$

see Fig. 2b. Above, $V'$ is the operator on the bath that appears in the probe-bath interaction, $V'_{\alpha\beta}$ is the off-diagonal matrix element of $V'$ between the eigenstates $|\mathcal{E}_\alpha\rangle$ and $|\mathcal{E}_\beta\rangle$ of the spin-bath system at coupling $J$, and $[\cdot]$ denotes an appropriate average over $\alpha, \beta$ within small energy windows.

Our primary analytical tool in the characterisation of the spin-bath system are the distribution of the fidelity susceptibility. The fidelity susceptibility $\chi_\alpha$ of an initial spin-bath product state $|\mathcal{E}_\alpha^0\rangle = |\sigma\rangle|E_a\rangle$ quantifies the first-order correction when a weak spin-bath coupling is switched on

$$\chi_\alpha = \langle \partial_J \mathcal{E}_\alpha | \partial_J \mathcal{E}_\alpha \rangle|_{J=0}. \tag{3}$$

The distribution of fidelity susceptibilities $f_{\mathrm{FS}}(\chi)$ is determined by the spectral properties of the bath alone. In Sec. 3, we compute the exact distribution $f_{\mathrm{FS}}$ of several Poisson random matrix ensembles, and for the Gaussian unitary ensemble. For the Gaussian orthogonal, Gaussian symplectic and ETH cases, we obtain exact forms for the asymptotes of $f_{\mathrm{FS}}$, and numerically exact forms for the full distribution.

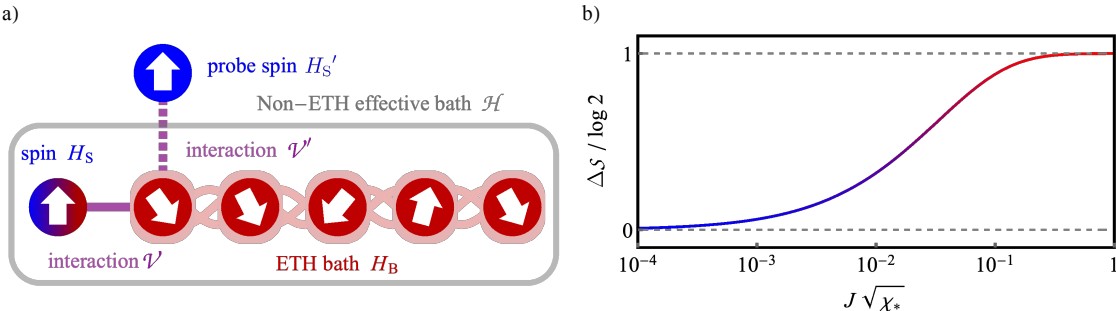

Figure 2: *The Spin-ETH model as an ETH-like bath*: a) at intermediate coupling, the Spin-ETH model appears as an effective bath to a second 'probe' spin. b) The entropy of the Spin-ETH model is enhanced from zero to $\log 2$ as the coupling to the first spin is tuned in the range $1/d \lesssim J\sqrt{\chi_\star} \ll 1$.

The distribution of fidelity susceptibilities $f_{FS}$ has several universal features. One feature that is central to our analysis is its heavy tail,

$$f_{FS}(\chi) \sim \sqrt{\frac{\chi_\star}{\chi^3}}. \tag{4}$$

The coefficient $\chi_\star$ sets the typical value. For random matrix and ETH baths (Sec. 3), $J^2 \chi_\star$ is equal to $g^2$ up to an $O(1)$ constant $c_\beta$ that depends on the symmetry class of the bath

$$J^2 \chi_\star = c_\beta g^2. \tag{5}$$

More broadly, as the heavy tail is a consequence of near degeneracies in the uncoupled many-body spectrum, Eq. (4) holds even if the bath does not satisfy ETH[2], and the dimensionless parameter $J\sqrt{\chi_\star}$ identifies the relevant reduced coupling. We use $J\sqrt{\chi_\star}$ as the reduced coupling henceforth.

States that contribute to the heavy tail of $f_{FS}$ are resonant with $O(1)$ other product states. We treat these resonances within a *two level resonant model* to obtain simple 'cat-state' ansatz for these states (Sec. 4). Several analytical results follow, specifically: (i) the universal shape of the spin entanglement entropy in eigenstates (Sec. 4.1), characterised by mean and typical entropies

$$S_{mean} = 2\pi J \sqrt{\chi_\star} + \cdots \tag{6a}$$

$$S_{median} = -c_{m.} J^2 \chi_\star \log c_{m.} J^2 \chi_\star + \cdots \tag{6b}$$

(here $c_{m.}$ is an $O(1)$ numerical constant), (ii) the infinite time-averaged spin-spin correlation function (Sec. 5)

$$\overline{\langle \sigma_P^z(t) \sigma_P^z(0) \rangle} = 1 - 4\pi J \sqrt{\frac{\chi_\star(0, h_S)}{6}} + \cdots \tag{7}$$

and (iii) the intermediate enhancement of the bath entropy (Sec. 7)

$$\Delta S = -8J \sqrt{\chi_\star} \log(J \sqrt{\chi_\star}) + \cdots \tag{8}$$

(where in each case ... indicates the presence of corrections which are sub-leading for $J\sqrt{\chi_\star} < 1$).

## 2 Model

We consider a partially thermalising system that is composed of a single spin-$\frac{1}{2}$ (S) that is weakly coupled by $\mathcal{V}$ to a thermal bath (B)

$$\mathcal{H} = \mathcal{H}_0 + \mathcal{V}. \tag{9}$$

Above $\mathcal{H}_0$, the Hamiltonian in the absence of S-E interactions, is given by,

$$\mathcal{H}_0 = H_S \otimes \mathbb{1} + \mathbb{1} \otimes H_B, \tag{10}$$

where $H_S$ is single spin-$\frac{1}{2}$ with level splitting $h_S$

$$H_S = \tfrac{1}{2} h_S \sigma_S^z, \tag{11}$$

---

[2]Indeed, Eq. (4) holds for an ensemble of many-body localised systems.

and $H_\text{B}$ is the Hamiltonian of a finite many-body quantum bath with density of states $\rho_0$ and dimension $d$ (we use calligraphic letters to denote global operators, and roman letters to denote those local to the system or bath). See Fig. 1a.

We focus on two classes of well-thermalising baths: (i) random baths with Hamiltonians drawn from Haar invariant random matrix ensembles (the *Spin-RM model*), and (ii) a spin chain with local interactions which satisfies ETH (the *Spin-ETH model*). We describe these in turn below.

At several points we will consider eigenstate averaged properties of mid spectrum states. When numerically evaluating these properties, the average is performed over the middle 25% of the spectrum obtained from exact diagonalisation.

## 2.1 Random matrix baths

In the Spin-RM model we consider six ensembles of random matrices: the three standard Gaussian random ensembles (GRE), and three ensembles with the same symmetries, but which lack level repulsion.

For the GRE case we take

$$H_\text{B} \sim \text{GOE}(d),\ \text{or}\ \text{GUE}(d),\ \text{or}\ \text{GSE}(d) \tag{12}$$

to be a $d \times d$ Gaussian random matrix of either real, complex and quaternionic elements (with Dyson indices $\beta = 1, 2, 4$ respectively). These distributions are extensively studied, see e.g. Ref. [35]. The matrix elements of $H_\text{B}$ are determined by the one and two point correlations

$$
\begin{aligned}
&[H_{\text{B},ij}] = 0\,, \\
&[H_{\text{B},ij} H_{\text{B},kl}{}^*] = \frac{1}{d}\,\delta_{ik}\delta_{jl} + \frac{2-\beta}{d\beta}\,\delta_{il}\delta_{jk}\,,
\end{aligned}
\tag{13}
$$

where $[\cdot]$ denotes ensemble averaging. The eigenvalues $H_\text{B}|E_a\rangle = E_a|E_a\rangle$ have mean and variance

$$[E_a] = 0\,, \tag{14a}$$

$$[E_a^2] = \frac{1}{d}\Big[\text{tr}\big(H_\text{B} H_\text{B}^\dagger\big)\Big] = 1 + O(d^{-1})\,. \tag{14b}$$

More precisely, the density of states of the bath is set by the Wigner semi-circle law

$$\rho(E) = \rho_0 \sqrt{1 - \frac{E^2}{4}} + O(d^{-1})\,, \tag{15}$$

with density of states at maximum entropy $\rho_0 = d/\pi$.

Throughout we assume that the dimension of the bath is large ($d \gg 1$), so that the mean energy level spacing of the bath is much smaller than the splitting $h_\text{S}$ of the spin energy levels, which is in turn smaller than the bandwidth of the bath

$$\rho_0^{-1} \ll h_\text{S} \ll \sqrt{[E_a^2]}\,. \tag{16}$$

Eq. (16) holds for a locally interacting many-body quantum bath with $L \gg 1$ degrees of freedom (the bandwidth grows asymptotically as $\sqrt{[E_a^2]} \propto \sqrt{L}$ and the density of states grows as $\log \rho_0 \propto L$).

We additionally define three "Poisson" ensembles with the same symmetries as the GRE, but which lack their characteristic level repulsion. These ensembles are of interest as we find

similar results as in the GRE, but the calculations are significantly more tractable. Specifically, we take

$$H_{\mathrm{B}} = U \Lambda U^{\dagger}, \tag{17}$$

where $\Lambda$ is a diagonal matrix with independent and identically distributed (iid) elements $E_a$ drawn from the semi-circle distribution (15), and $U$ drawn from the Haar invariant ensemble of $d \times d$ unitary matrices with elements that are either real ($U \sim \mathrm{CRE}(d)$, the circular real ensemble), complex ($U \sim \mathrm{CUE}(d)$, the circular unitary ensemble) or quaternionic ($U \sim \mathrm{CQE}(d)$, the circular quaternionic ensemble). We refer to these distributions as P×CRE, P×CUE, and P×CQE respectively. This construction yields ensembles of matrices with Poissonian level statistics, but with the (i) same density of states (15), (ii) same marginal distribution of matrix elements at large $d$, (iii) same symmetries, and (iv) same Haar invariance as GOE($d$), GUE($d$), and GSE($d$) respectively.

We ascribe the distributions P×CRE, P×CUE, and P×CQE indices $\beta = 1, 2, 4$ respectively. This labelling differs from the standard one of $\beta = 0$ in random matrix theory because the marginal distribution of the matrix elements is the only relevant quantity here. Specifically, in the limit of large $d$, the marginal distribution of the matrix elements for Poissonian $H_{\mathrm{B}}$ is Gaussian with zero mean and the same two point correlations as the equivalent GRE.

## 2.2 A many-body quantum system as a bath

In the *Spin-ETH model*, the bath is a thermalising many-body quantum system with local interactions. Specifically, we choose $H_{\mathrm{B}}$ to describe a weakly disordered Ising model with longitudinal and transverse fields

$$H_{\mathrm{B}} = \sum_{n=1}^{L} \left( (-1)^n \sigma_n^x \sigma_{n+1}^x + h_n \sigma_n^x + u \Gamma \sigma_n^z \right), \tag{18}$$

with open boundary conditions $\sigma_{L+1}^z = 0$. The longitudinal fields $h_n$ are iid random variables drawn from a uniform distribution with mean $[h_n] = h$ and variance $[h_n^2] - [h_n]^2 = u^2(1 - \Gamma^2)$. Following Refs. [30, 36] we set

$$(h, u, \Gamma) = (0.8090, 0.9045, 0.9950). \tag{19}$$

The weak disorder breaks the inversion symmetry of the system, while the small disorder bandwidth, $|h_n - h| \leq \delta h$ with $\delta h = u \sqrt{3(1 - \Gamma^2)} \approx 0.14$, is well below the interaction energy scale ensuring that there are no presages to localisation.

The alternating ferromagnetic and anti-ferromagnetic couplings ensure that the density of states $\rho(E)$ is Gaussian at small system sizes, and independent of the choice of $h, u, \Gamma, L$ (in contrast, the density of states has a marked asymmetry at accessible systems sizes for homogeneous couplings). Specifically, $H_{\mathrm{B}}$ has density of states

$$\rho(E) = \rho_0 e^{-E^2/(2s_E^2)}, \tag{20}$$

with mean $[E_a] = 0$ and variance $[E_a^2] = s_E^2 = \mathrm{tr}\left(H_{\mathrm{B}}^2\right)/2^L = L(1 + u^2 + h^2)$). The Hilbert space dimension dimension and density of states at maximum entropy are given by

$$d = 2^L, \qquad \rho_0 = \frac{d}{\sqrt{2\pi s_E^2}}. \tag{21}$$

Throughout this manuscript, when considering the Spin-ETH model, we set the probe spin field to

$$h_{\mathrm{S}} = \sqrt{h^2 + u^2} \approx 1.21, \tag{22}$$

so that the probe field is half the value of a typical local field in the Ising chain.

### 2.3 Spin-Bath interactions

Throughout our analysis, the interaction may be considered to be generic,

$$\mathcal{V} = J\left(\sigma_S^+ \otimes V^\dagger + \sigma_S^- \otimes V\right) + J_z \sigma_S^z \otimes V', \tag{23}$$

where $J, J_z > 0$ are coupling constants of comparable size $J_z = O(J)$, $\sigma_S^\pm$ and $\sigma_S^z$ are the usual Pauli matrices on the spin, and $V, V'$ are operators on the bath with $\mathrm{tr}\left(VV^\dagger\right) = \mathrm{tr}\left(V'V'\right) = d$.

For the purposes of specificity, in numerics, we choose $V = V^\dagger$, $V' = 0$ to yield

$$\mathcal{V} = J\sigma_S^x \otimes V. \tag{24}$$

In the Spin-RM model we set $V$ to be the diagonal matrix $V_{ij} = \delta_{ij}(-1)^j$. In the Spin-ETH model we choose set $V = \sigma_m^x$ where $m$ is the mid-chain site $m = \lfloor (L+1)/2 \rfloor$.

## 3 Weak coupling: $J\sqrt{\chi_\star} \ll 1/d$

The late time properties of dynamical evolution are captured by the system's steady states: the eigenstates $|\mathcal{E}_\alpha\rangle$. In the weak coupling limit, we characterise each $|\mathcal{E}_\alpha\rangle$ by a single quantity, its associated *fidelity susceptibility* $\chi_\alpha$. We subsequently obtain a statistical description of the $\chi_\alpha$ across eigenstates. In the weak coupling regime this may be used directly to obtain the distribution of spin entanglement entropies across eigenstates in the Spin-RM and Spin-ETH models.

### 3.1 The fidelity susceptibility

The change to each eigenstate upon deviating away from zero coupling is captured by its fidelity susceptibility. At zero coupling, the eigenstates are simple product states of the spin and bath

$$|\mathcal{E}_\alpha^0\rangle = |\sigma\rangle|E_a\rangle, \tag{25}$$

where $\alpha = (\sigma, a)$, $\sigma \in \{\uparrow, \downarrow\}$, with associated energies

$$\mathcal{E}_\alpha^0 = \tfrac{1}{2}\sigma h_S + E_a, \tag{26}$$

using $\uparrow = +1$ and $\downarrow = -1$. For small $J, J_z$, corrections to the decoupled limit may be obtained in perturbation theory

$$|\mathcal{E}_\alpha\rangle = |\mathcal{E}_\alpha^0\rangle + J|\partial_J \mathcal{E}_\alpha\rangle + J_z|\partial_{J_z} \mathcal{E}_\alpha\rangle + \dots. \tag{27}$$

We may associate a *fidelity susceptibility* $\chi_\alpha$ to each state, given by the squared norm of the first order correction in $J$

$$\chi_\alpha := \langle \partial_J \mathcal{E}_\alpha | \partial_J \mathcal{E}_\alpha \rangle = \sum_b \left| \frac{V_{ab}}{E_a - E_b + \sigma h_S} \right|^2. \tag{28}$$

Here $V_{ab} = \langle E_a|V|E_b\rangle$ are the matrix elements of the coupling operator $V$.

When $J \neq 0$, the eigenstates are entangled states of the spin and bath. The von Neumann entropy of the spin quantifies the entanglement between the spin and bath,

$$S_\alpha := -\mathrm{tr}(\hat{\rho}_\alpha \log \hat{\rho}_\alpha), \tag{29}$$

where $\hat{\rho}_\alpha$ is the reduced density matrix of the spin obtained from the eigenstate $|\mathcal{E}_\alpha\rangle$. For typical states, we obtain the entropy by expanding $\hat{\rho}_\alpha$ to leading order,

$$\hat{\rho}_\alpha = \begin{pmatrix} 1 - J^2\chi_\alpha & O\left(\frac{g}{\rho_0 h_\mathrm{S}}\right) \\ O\left(\frac{g}{\rho_0 h_\mathrm{S}}\right) & J^2\chi_\alpha \end{pmatrix} + O\left(\frac{g^2}{\rho_0 h_\mathrm{S}}\right) + O(g^3), \tag{30}$$

which yields

$$S_\alpha = J^2\chi_\alpha(1 - \log(J^2\chi_\alpha)) + O\left(\frac{g^2}{\rho_0 h_\mathrm{S}}\right) + O\left(g^3\right), \tag{31}$$

where $g$ is the reduced coupling (1a). Eqs. (30) and (31) are obtained in Appendix A by expanding to leading order in two small parameters: (i) the reduced coupling $g$, and (ii) the ratio of level spacings to field strengths $(\rho_0 h_\mathrm{S})^{-1} = O(1/d)$. This provides the leading order entanglement entropy, which is found to depend on $J$ but not $J_z$. Intuitively, this is because this term generates hybridisation between states in the same spin sector and leaves the reduced density matrix of the spin unaltered.

As (31) holds only in the perturbative limit $J^2\chi_\alpha \ll 1$, it is useful to estimate the scale of $\chi_\alpha$. For typical states we find that $J^2\chi_\alpha = O(g^2)$. This is seen by noting that $\chi_\alpha$ is dominated by the terms in the sum (28) with the smallest denominators $\min_b |E_a - E_b + h_\mathrm{S}| \approx 1/\rho_0$, whereas typical matrix elements are of size $|V_{ab}| \approx \sqrt{\mathrm{tr}(VV^\dagger)}/d = 1/\sqrt{d}$. Combining these estimates with (28) we obtain

$$J^2\chi_\mathrm{typ.} \approx \left(\frac{J\rho_0}{\sqrt{d}}\right)^2 = g^2. \tag{32}$$

Eq. (32) describes typical values as defined by the median $\chi_\mathrm{typ.} = \mathrm{med}_\alpha\, \chi_\alpha$, or the geometric mean $\chi_\mathrm{typ.} = \exp[\log \chi_\alpha]$. However, we will see that the fidelity susceptibility $\chi_\alpha$ is broadly distributed with no convergent arithmetic mean. As a result, $\chi_\mathrm{typ.}$ does not provide a satisfactory characterisation of the distribution of values $\chi_\alpha$ which we calculate in Sec. 3.2.

The fidelity susceptibility $\chi_\alpha$ is a well known quantity, most often studied as a probe of ground state phase transitions (see e.g. Refs. [37,38]). Recently, $\chi_\alpha$ and closely related quantities have been studied for mid-spectrum states in the context of quantum chaos [39–44]. The fidelity susceptibility is named for its appearance when the fidelity between the eigenstates of $\mathcal{H}$ and $\mathcal{H}_0$

$$F_\alpha(J, J_z) := |\langle \mathcal{E}_\alpha | \mathcal{E}_\alpha^0 \rangle| \tag{33}$$

is expanded in powers of the coupling $J$. In this case, when $J_z = 0$, $\chi_\alpha$ sets the leading order correction

$$F_\alpha(J, 0) = 1 - \tfrac{1}{2}J^2\chi_\alpha + O(J^4). \tag{34}$$

As the spin-bath coupling is determined by two parameters, $J$ and $J_z$, similar susceptibilities may be defined for the quadratic $JJ_z$ and $J_z^2$ terms in the expansion of $F_\alpha(J, J_z)$. However, as these terms do not contribute to the eigenstate entanglement of the spin, they are not of interest in the present context.

The quantity $\chi_\alpha$ is also relates closely to previous studies of the in the context of localisation. Operator expectation values on the spin can be calculated to leading order in the coupling $J$ using (30). This relates closely to the locator expansion in which local expectation values in an extended system are expanded in the inter-site coupling. This was considered in Anderson's seminal analysis of a localised lattice system [45] and in many subsequent works, for example [46–49].

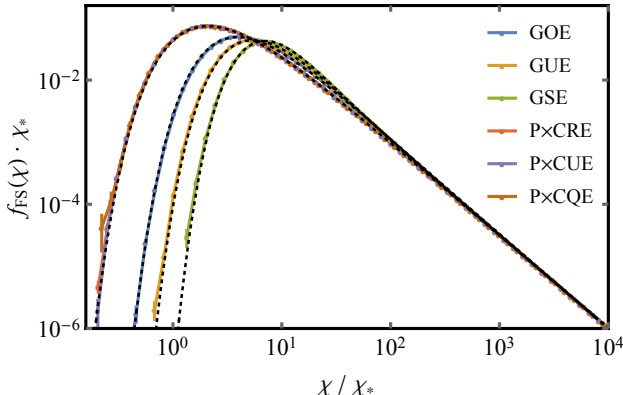

Figure 3: *The distribution $f_{\text{FS}}(\chi)$ of the fidelity susceptibility in random matrix theory ensembles:* Numerically calculated distributions of the fidelity susceptibility (solid colours, error bars indicate 68% confidence interval) are compared with analytic predictions (black, dotted). The numerical distributions are obtained by histogramming the fidelity susceptibility (28) of mid-spectrum states obtained from exact diagonalisation. In each case the distribution has been re-scaled by the maximum entropy value $\chi_\star(0, h_{\text{S}})$. The dotted curves have no fitted parameters in the case of the Poisson (see (39)) and GUE (see (51a)) ensembles. For GOE and GSE the dashed line has the exact limiting behaviour given by (47) and (49) whereas values of $f_{\text{FS}}$ at intermediate values of $\chi$ is obtained by a one parameter fit (details in text). Parameters: $h_{\text{S}} = 0.1$, $d = 2048$, $N = 3000$ realisations for GSE and P×CQE, $N = 10^5$ realisations otherwise.

## 3.2 The distribution $f_{\text{FS}}(\chi)$ in Haar invariant random matrix ensembles

In the weak coupling limit we have a one-to-one relationship between the fidelity susceptibility $\chi_\alpha$, and the entanglement entropy $S_\alpha$ (31). Thus, to obtain the distribution of entanglement entropies, a quantity of physical interest, it is sufficient to calculate the distribution of $\chi_\alpha$. In this section we calculate the distribution of the fidelity susceptibility $\chi_\alpha$ of the state $\alpha = (\sigma, a)$, obtained by ensemble averaging

$$f_{\text{FS}}(\chi | E, \sigma h) := \frac{[\delta(\chi - \chi_\alpha)\delta(E - E_a)]}{[\delta(E - E_a)]} . \tag{35}$$

This distribution carries two dependencies: the initial energy $E$ of the bath, and $\sigma h_{\text{S}}$ the energy transferred into the bath in order to flip the spin. We perform this calculation for the Haar invariant ensembles of Sec. 2.1 in the limit of large bath dimension $d$. We confirm this calculation with numerics for finite $d$ (Fig. 3).

### 3.2.1 $f_{\text{FS}}(\chi)$ for Haar random baths with Poisson level statistics

We begin with the simplest case, where $H_{\text{B}}$ is a Haar random matrix with Poisson level statistics. We obtain an explicit form for $f_{\text{FS}}$ before discussing the key features of the distribution.

We first consider the cumulant generating function

$$\begin{aligned} K(t | E, \sigma h_{\text{S}}) &:= \log\left(\int \mathrm{d}\chi \, e^{it\chi/d} f_{\text{FS}}(\chi | E, \sigma h_{\text{S}})\right) \\ &= \log\frac{[e^{it\chi_\alpha/d}\delta(E - E_a)]}{[\delta(E - E_a)]} \end{aligned} \tag{36}$$

and substitute in the definition of $\chi_\alpha$ to obtain

$$K(t|E,\omega) = d\log\left[\exp\left(\frac{\mathrm{i}t}{d}\left|\frac{V_{ab}}{E-E_b+\omega}\right|^2\right)\right].\tag{37}$$

In the Poisson case, at large $d$, we may treat each matrix element $V_{ab}$ and each energy level $E_a$ as iid random variables. The ensemble averaging is then straightforward (see Appendix B) and yields

$$\lim_{d\to\infty} K(t|E,\omega) = -\sqrt{-\frac{4\pi\mathrm{i}t\rho(E+\omega)^2[|V_{ab}|]^2}{d}}\,.\tag{38}$$

Inverting the relation (36) we obtain a Levy distribution

$$f_{\mathrm{FS}}(\chi|E,\omega) = \exp\left(-\pi\frac{\chi_\star(E,\omega)}{\chi}\right)\sqrt{\frac{\chi_\star(E,\omega)}{\chi^3}}\,,\tag{39}$$

with a characteristic scale set by

$$\chi_\star(E,\omega) = [|V_{ab}|]^2\rho(E+\omega)^2\,.\tag{40}$$

$\chi_\star(E,\omega)$ sets the typical values of $\chi_\alpha$. It is the scale obtained from the definition of $\chi_\alpha$ (28), by approximating the sum with its dominant term, and replacing the numerator and denominator with their typical values $[|V_{ab}|]^2$ and $\rho(E+h_S)^{-2}$ respectively.

We note that the Levy distribution may be related to the more familiar normal distribution. Precisely, $\chi$ has the same distribution as $2\pi\chi_\star/z^2$ for $z$ drawn from the standard normal distribution $z\sim\mathcal{N}\left(\mu=0,\sigma^2=1\right)$.

Further calculation relates $\chi_\star(E,\omega)$ to the parameters of the Spin-RM model. Specifically, we use that the matrix elements $V_{ab}$ converge on a Gaussian distribution with mean $[V_{ab}]=0$ and variance $[|V_{ab}|^2]=1/d$. Thus the distribution of the absolute value $|V_{ab}|$ of the matrix elements has distribution

$$f_{\mathrm{ME}}(|V_{ab}|) \propto |V_{ab}|^{\beta-1}\exp\left(-\tfrac{1}{2}d\beta|V_{ab}|^2\right)\tag{41}$$

and hence a mean

$$[|V_{ab}|] = \sqrt{\frac{2}{d\beta}}\cdot\frac{\Gamma(\frac{1+\beta}{2})}{\Gamma(\frac{\beta}{2})} =: \sqrt{\frac{c_\beta}{d}}\,.\tag{42}$$

In (42) $\Gamma(\cdot)$ is the gamma function, the Dyson index $\beta=1,2,4$ for real, complex and quaternionic matrix elements respectively, and we have defined the numerical constant $c_\beta$ whose value depends only on the symmetry class of the matrix

$$c_\beta = \begin{cases} 2/\pi & \beta=1 & (\text{GOE, P}\times\text{CRE}),\\ \pi/4 & \text{for} \quad \beta=2 & (\text{GUE, P}\times\text{CUE}),\\ 9\pi/32 & \beta=4 & (\text{GSE, P}\times\text{CQE}). \end{cases}\tag{43}$$

Thus, in terms of the bare properties of the Poissonian bath, we have explicit forms for both the distribution $f_{\mathrm{FS}}$ (39) and its typical values $\chi_\star$

$$\chi_\star(E,\omega) = c_\beta\frac{\rho(E+\omega)^2}{d}\,.\tag{44}$$

In Fig 3 we compare these predictions with numerics. Eq. (40) is the left-most black-dashed curve plotted in Fig 3. This curve shows good agreement with the corresponding numerically calculated fidelity susceptibility distributions for the Spin-RM model for P×CRE, P×CUE, and P×CQE baths (the red, purple and brown curves respectively which lie on top of each other).

### 3.2.2 $f_{FS}$ for other random matrix baths

We highlight three features of $f_{FS}$, as calculated for the Poisson case (39). These feature of $f_{FS}$ found for any choice of thermal bath $H_B$:

i) The heavy tail of the distribution, decaying as $f_{FS} \sim \chi_\star^{1/2}/\chi^{3/2}$ leads to rare, large values of $\chi_\alpha$ and prevents the convergence of the arithmetic mean.

ii) The rapid decay at small $\chi \lesssim \chi_\star$ is faster than any power law, to leading order $\log f_{FS} \propto -\chi^{-1}$.

iii) The scale of typical values $\chi_\alpha$ is set by $\chi_\star$.

Elaborating on these points:

*i) Rare large values*: The large values of $\chi_\alpha$ correspond to states where there is an unexpectedly close resonance which dominates the sum in (28). The effect of such close many-body resonances gives rise to the $\chi^{-3/2}$ tail irrespective of the choice of random ensemble $H_B$. To see this, let us approximate

$$\chi_\alpha \approx \left| \frac{V_{ab}}{E_a - E_b + \sigma h_S} \right|^2 , \tag{45}$$

where in each case $b$ is chosen to minimise the denominator. We then write $f_{LS}(\Delta_{ab})$ for the distribution of the energy separation to the nearest level $\Delta_{ab} = |E_a - E_b + \sigma h_S|$ in the opposite spin sector, and, as before, $f_{ME}$ for the distribution of matrix elements $|V_{ab}|$. Within this approximation, (which becomes exact for asymptotically large $\chi$)

$$
\begin{aligned}
f_{FS} &= \int_0^\infty dV \int_0^\infty d\Delta\, \delta\left( \chi - \left|\frac{V}{\Delta}\right|^2 \right) f_{ME}(V) f_{LS}(\Delta) \\
&= \frac{1}{2\chi^{3/2}} \int_0^\infty dV |V| f_{ME}(V) f_{LS}\left( \frac{V}{\sqrt{\chi}} \right).
\end{aligned}
\tag{46}
$$

The asymptotic behaviour

$$f_{FS}(\chi|E,\omega) \sim \sqrt{\frac{\chi_\star(E,\omega)}{\chi^3}} \tag{47}$$

then follows from taking the limit

$$
\begin{aligned}
\chi_\star(E,\omega) &= \lim_{\chi\to\infty} \chi^3 f_{FS}^2(\chi) \\
&= \left( \frac{1}{2} \int dV |V| f_{ME}(V) f_{LS}(0) \right)^2 \\
&= [|V|]^2 \rho(E+\omega)^2.
\end{aligned}
\tag{48}
$$

Here we have set $E = E_a$ and $\omega = \sigma h_S$. We have also used that $\lim_{\Delta\to 0} f_{LS}(\Delta) = 2\rho(E)$, which holds irrespective of the level statistics with an sector. Note that (48) is in exact agreement with (40). Naively one might expect to obtain different asymptotic behaviours for $f_{FS}$ depending on the level statistics of the bath matrix, as the values of $E_a, E_b$ will be correlated. That we recover the same form independent of the bath matrix ensemble follows from the fact that the shift by $\sigma h_S$ conceals the correlated nature of the energy levels. This point has been previously noted in studies of the locator expansion in the context of localisation. [46, 48, 49]

*ii) Fast decay at small $\chi$*: Below the scale of the typical fidelity susceptibility $\chi \lesssim \chi_{typ.}$ the distribution converges very quickly to zero $\log f_{FS} \propto -\chi^{-1} + O(\log\chi)$.

In the Poisson case, the strong suppression of $f_{FS}$ at small $\chi$ reflects that atypically small values of $\chi_\alpha$ occur only when each of the iid terms in the sum $\chi_\alpha$ (28) is independently small.

Small values of $\chi$ occur because large numbers of the matrix elements $V_{ab}$ are atypically small, or because large numbers of the energy levels are atypically far from $E_a + \sigma h_S$.

For the GRE baths the terms in $\chi_\alpha$ are not mutually independent. Instead, spectral rigidity suppresses the fluctuations on the energy levels so that small $\chi_\alpha$ values occur only due to small matrix elements. This distinction in the GRE leads only to an $O(1)$ quantitative change to the small $\chi$ behaviour

$$\log f_{\mathrm{FS}}(\chi|E,\omega) \sim -\frac{\chi_\star(E,\omega)}{\chi} \times \begin{cases} \pi & \text{Poisson}, \\ \dfrac{\beta \pi^2}{2c_\beta} & \text{GRE}. \end{cases} \tag{49}$$

We show how (49) is obtained in Sec. 3.2.3.

*iii) Typical value of $\chi_\alpha$:* The scale of typical values $\chi$ is set by the peak of the distribution and unaffected by the heavy tail. Specifically, the geometric mean is given by

$$\chi_{\mathrm{typ.}}(E,\omega) = \exp\left(\int \mathrm{d}\chi\, f_{\mathrm{FS}}(\chi|E,\omega) \log\chi\right)$$
$$= c_{\mathrm{typ.}}\,\chi_\star(E,\omega), \tag{50}$$

where $c_{\mathrm{typ.}} = O(1)$ is a numerical constant. For example, in the Poisson ensembles this constant has value $c_{\mathrm{typ.}} = 4\pi e^\gamma$ where $\gamma = 0.57721...$ is the Euler-Mascheroni constant.

### 3.2.3 $f_{\mathrm{FS}}(\chi)$ for Gaussian random matrix baths

We extend our analysis to obtain forms for the distribution of fidelity susceptibilities $f_{\mathrm{FS}}$ for $H_{\mathrm{B}}$ drawn from one of the GRE ensembles. This extension is desirable as GRE matrices predict the eigenstate properties of thermalising many body quantum systems.

In Appendix C we calculate $f_{\mathrm{FS}}$ exactly for a GUE ensemble ($\beta = 2$)

$$f_{\mathrm{FS}}^{\mathrm{GUE}}(\chi) = \exp\left(-\frac{4\pi\chi_\star}{\chi}\right)\sqrt{\frac{\chi_\star}{\chi^3}}\left(1 + \frac{8\pi\chi_\star}{\chi}\right). \tag{51a}$$

Above, we suppress the $(E,\omega)$ dependency of $f_{\mathrm{FS}}$ and $\chi_\star$ for brevity. We further calculate $f_{\mathrm{FS}}$ for the GOE ($\beta = 1$) or GSE ($\beta = 4$) cases up to some undetermined numerical constants $(C_{1,2}, C'_{1,2})$

$$f_{\mathrm{FS}}^{\mathrm{GOE}} = \exp\left(-\frac{\pi^3\chi_\star}{4\chi}\right)\sqrt{\frac{\chi_\star}{\chi^3}}\left(1 + C_1\sqrt{\frac{\chi_\star}{\chi}} + C_2\frac{\chi_\star}{\chi} + O\left(\frac{\chi_\star}{\chi}\right)^{3/2}\right), \tag{51b}$$

$$f_{\mathrm{FS}}^{\mathrm{GSE}} = \exp\left(-\frac{9\pi\chi_\star}{64\chi}\right)\sqrt{\frac{\chi_\star}{\chi^3}}\left(1 + C'_1\frac{\chi_\star}{\chi} + C'_2\frac{\chi_\star^2}{\chi^2}\right). \tag{51c}$$

The number of undetermined parameters is reduced by enforcing the normalisation condition $\int \mathrm{d}\chi\, f_{\mathrm{FS}}(\chi) = 1$:

$$4\pi C_1 + 4C_2 - (\pi - 2)\pi^3 = 0,$$
$$8192 C'_1 + 768\pi C'_2 + 135\pi^2 = 0, \tag{52}$$

where we have neglected sub-leading $O(\chi_\star/\chi)^{3/2}$ corrections in the GOE case. Throughout the rest of this paper we use the GOE values $C_1 = 5.29...$, $C_2 = 11.19...$ determined by a least square numerical fit.

We compare (51) with numerics in Figure 3. As with the Poisson case, $f_{\mathrm{FS}}$ is numerically calculated by averaging over the mid-spectrum states for $h_S = 0.1$ and $d = 2048$. In each

case there is convincing agreement between the analytic forms (black, dotted) and numerical calculations (solid colours). These analytic forms are specified with no free parameters in the case of Poisson (39) and GUE (51a). In the case of GOE and GSE the parameters $C_2$, $C_2'$ are fixed by the normalisation condition (52), whereas the remaining free parameters $C_1$, $C_1'$ are determined by a one-parameter least squares fit. For this numerical analysis we neglect the sub-leading $O(\chi_\star/\chi)^{3/2}$ corrections in the GOE case.

The full derivation of (51) (Appendix C) is involved, however the asymptotic forms may be derived in a few lines. The large $\chi$ form is obtained exactly as in (47). The the small $\chi$ form, given by (49), we obtain here. We start from the definition of the cumulant generating function (36). In the GRE case, for $d \gg 1$, the matrix elements may be treated as iid drawn from the distribution (41). The corrections resulting from this approximation are $O(1/d)$ [50–52], and we neglect them throughout this section. Thus, integrating over the matrix elements yields

$$
\begin{aligned}
K(t|E,0) &= \log\left[\exp\left(\frac{it}{d}\sum_b\left|\frac{V_{ab}}{E-E_b}\right|^2\right)\right]_{V_{ab},E_b}\\
&= \log\left[\prod_b\left(1-\frac{2it[|V_{ab}^2|]}{\beta d|E-E_b|^2}\right)^{-\beta/2}\right]_{E_b}.
\end{aligned}
\tag{53}
$$

We use the identity $\log\prod_b g(E_b) = \sum_b \log g(E_b)$ to replace the sum over levels with an integration over the ensemble averaged density of states $\sum_b \log g(E_b) \to \int dE' \rho(E') \log g(E') + O(1/d)$

$$
K(t|E,0) = -\frac{\beta}{2}\int dE'\rho(E')\log\left(1-\frac{2it[|V_{ab}^2|]}{\beta d|E-E'|^2}\right).
\tag{54}
$$

This replacement is only valid if the density of states is smooth on the scale on which the summand in (53) varies. That is, if the width of the peak of the summand is much greater than the level spacing. Note (i) the summand has a single peak with a width $\Delta E \approx \sqrt{2t[|V_{ab}^2|]/\beta d}$ (where $[|V_{ab}^2|] = 1/d$); (ii) the level spacing is on a scale $\rho(E)^{-1} \propto d^{-1}$. Thus the sum-to-integral replacement is valid in the limit $t \gg 1$. The integral may be further simplified by assuming the peak of the integrand is much narrower than the bandwidth (requiring $t \ll d^2$). In this limit the integrand is sharply peaked at $E' \approx E$ allowing use to substitute $\rho(E') \to \rho(E)$ and integrate

$$
K(t|E,0) \sim -\sqrt{-\frac{2\pi^2 it\beta\rho(E)^2[|V_{ab}^2|]}{d}} \quad (1 \ll t \ll d^2).
\tag{55}
$$

As the large $t$ behaviour of $K(t|E,0)$ sets the small $\chi$ behaviour of $f_{\text{FS}}$, by inverting the Fourier transform we obtain the low $\chi$ asymptote

$$
-\log f_{\text{FS}}(\chi|E,0) \sim \frac{\pi^2\beta\rho(E)^2[|V_{ab}^2|]}{2\chi} = \frac{\beta\pi^2\chi_\star}{2c_\beta\chi},
\tag{56}
$$

where $\sim$ indicates asymptotic equality in the small $\chi$ limit. Combining this GRE result, with the Poisson result (39) we obtain (49). This shows that the lower tail is sensitive to both symmetry class, and level statistics.

We make a comment on the scope of this derivation. In obtaining (54), we replaced the density of states of $H_B$ with the ensemble averaged density of states. This replacement assumes fluctuations on the density of states are negligible. For Poisson level statistics this assumption is invalid, as samples in which $\rho(E+\omega)$ is atypically small make a significant contribution to the

lower tail, and thus (56) does not agree with the previously derived behaviour of Poissonian Spin-RM models (39). However, this replacement is justified for GRE matrices exhibit much smaller instance to instance variation on the density of states.

### 3.2.4 The distribution of $\chi_\alpha$ over states within a sample

The distribution $f_{\text{FS}}$ is self averaging. That is, in the limit of large $d$, the distributions obtained in this section hold for $\chi_\alpha$ obtained for states within a small energy window of a single Spin-RM Hamiltonian (specifically an energy window much smaller than the bandwidth, but much larger than the level spacing). Intuitively, the fidelity susceptibility of each state is dominated by its coupling to nearby states (which generate large terms in $\chi_\alpha$), and is uncorrelated with the properties of energetically distant states [50–52].

## 3.3 The distribution $f_{\text{FS}}$ in ETH systems

We extend our calculation of $f_{\text{FS}}$ to the more physical case of a bath that is a locally interacting, many body quantum system. Specifically, we use eigenstate thermalisation hypothesis (ETH) to adapt the GRE calculation of $f_{\text{FS}}$ (Sec. 3.2) to this setting, and numerically verify the predicted form of $f_{\text{FS}}$ in the Spin-ETH model.

### 3.3.1 Statement of ETH

ETH describes how isolated quantum systems approach an equilibrium described by quantum statistical mechanics [26–29] (for an overview see Ref. [31] and references therein). Let $H_{\text{B}}$ be a generic, locally interacting, thermalising quantum system. For specificity we assume $H_{\text{B}}$ to be a length $L$ chain of interacting spins-$\frac{1}{2}$, such as the Ising chain (18). ETH provides an ansatz for the matrix elements of a local operator $V$ evaluated in the eigenbasis of $H_{\text{B}}$

$$V_{ab} = \bar{V}(E_a)\,\delta_{ab} + \sqrt{\frac{\tilde{v}(E_a, E_b - E_a)}{\rho(E_b)}}\, R_{ab}\,, \tag{57}$$

where $R_{ab}$ are iid Gaussian random numbers with zero mean $[R_{ab}] = 0$ and unit variance $[|R_{ab}|^2] = 1$, $\bar{V}(E)$ and $\tilde{v}(E, \omega)$ are real functions smooth in their arguments, and $\tilde{v}(E, \omega)$ is non-negative. $\bar{V}(E)$ and $\tilde{v}(E, \omega)$ are further determined by physical considerations: Hermiticity enforces

$$\tilde{v}(E, \omega)\rho(E) = \tilde{v}(E + \omega, -\omega)\rho(E + \omega)\,, \tag{58}$$

while the one and two-time correlation functions evaluated in the micro-canonical ensemble are given by

$$\text{tr}(V\hat{\rho}_E) = \bar{V}(E)\,, \tag{59a}$$

$$\text{tr}\left(e^{iHt}V e^{-iHt} V\hat{\rho}_E\right) = \bar{V}(E)^2 + \int d\omega\, \tilde{v}(E, \omega)\,e^{i\omega t}\,, \tag{59b}$$

up to $O(1/d) = O(2^{-L})$ corrections. Here $\hat{\rho}_E$ is a micro-canonical ensemble of energy $E$ and window width $\Delta$

$$\hat{\rho}_E = \frac{1}{N_E} \sum_a \mathbf{1}_E(E_a)|E_a\rangle\langle E_a|\,, \tag{60}$$

where indicator function $\mathbf{1}_E(E_a)$ is given by

$$\mathbf{1}_E(E') := \begin{cases} 1 & |E - E'| < \Delta/2\,, \\ 0 & \text{otherwise}\,, \end{cases} \tag{61}$$

and $N_E := \sum_a \mathbf{1}_E(E_a)$ enforces normalisation. The micro-canonical window width $\Delta$ is chosen to be much smaller than the scale on which $\rho(E)$, $\bar{V}(E)$ or $\tilde{v}(E, \omega)$ vary, but much greater than level spacing

$$|\partial_E \tilde{v}(E, \omega)| = O\left(\frac{\tilde{v}(E, \omega)}{L}\right) \ll \Delta^{-1} \ll \rho(E). \tag{62}$$

### 3.3.2 The distribution $f_{\text{FS}}$

The GRE results (Sec. 3.2) are adapted to the ETH setting by repeating the derivations with the relationship

$$[|V_{ab}|]^2 = c_\beta [|V_{ab}|^2] = c_\beta \frac{\tilde{v}(E_a, E_b - E_a)}{\rho(E_b)}. \tag{63}$$

The resulting distributions are as in GRE case (51) but with a typical scale set by

$$\chi_\star(E, \omega) = c_\beta \tilde{v}(E, \omega)\rho(E + \omega). \tag{64}$$

The cases $\beta = 1, 2, 4$, (corresponding to $R_{ab} \in \mathbb{R}, \mathbb{C}, \mathbb{H}$) correspond naturally to the GOE, GUE and GSE ensembles. Physically these cases describe systems with time reversal symmetry $[\mathcal{T}, \mathcal{H}] = 0$ ($\beta = 1, 4$), or without ($\beta = 2$). The time reversal symmetric cases are distinguished by whether the anti-unitary time reversal symmetry operator squares to positive unity $\mathcal{T}^2 = 1$ ($\beta = 1$) or negative unity $\mathcal{T}^2 = -1$ ($\beta = 4$) [53].

In Fig 4 we numerically verify the form of $f_{\text{FS}}$ in the Spin-ETH model with $H_B$ given by the weakly disordered interacting Ising chain (18). The $\chi_\alpha$ are obtained from the mid-spectrum states of $N = 1000$ realisations with $h_S = \sqrt{h^2 + u^2} \approx 1.21$, and $V = \sigma_m^x$ for $m = \lfloor (L+1)/2 \rfloor$. The numerically calculated distribution (solid colours) agrees with the corresponding theoretical predictions (dashed colour) for all values of bath size $L$ (legend inset). The correct large $\chi$ behaviour (47) is observed for all $L$, whereas there is discrepancy at small $\chi$ between the data and prediction which is disappearing at large $L$. The small $\chi$ discrepancy is a finite size effect which causes the asymptotic $\log f_{\text{FS}} \sim -\chi_\star/\chi$ decay at small $\chi$ to be cut off by a slower

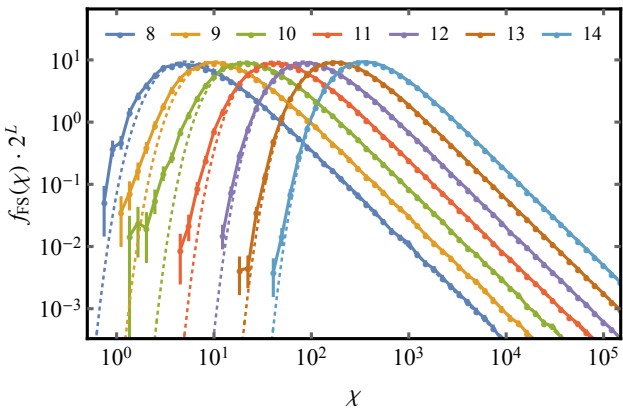

Figure 4: $f_{\text{FS}}$ *in thermalising quantum systems:* Numerically calculated distributions of $f_{\text{FS}}$ in the Spin-ETH model (solid points, colour) are compared with analytic predictions for an ETH system (dotted lines, colour). Each numerical distribution was produced by histogramming values of $\chi_\alpha$ obtained from the mid-spectrum states. Error-bars indicate standard error on the mean. The legend shows the bath sizes $L$, the other parameters are as in the main text. $N = 300$ realisations for $L = 14$, $N = 1000$ otherwise.

power law behaviour $f_{FS} \sim \chi^k$ with an exponent $k$ that grows exponentially in the system size $L$ (see Appendix C). The theory curves are given by (51b) with $\chi_\star(0, h_S)$ given by (64), and

$$\rho(E + \omega) = \left[\frac{1}{\Delta N_E} \sum_{ab} \mathbf{1}_E(E_a)\mathbf{1}_{E_a+\omega}(E_b)\right] + O(\Delta), \tag{65}$$

$$\tilde{\nu}(E, \omega) = \left[\frac{1}{\Delta N_E} \sum_{ab} \mathbf{1}_E(E_a)\mathbf{1}_{E_a+\omega}(E_b)|V_{ab}|^2\right] + O(\Delta), \tag{66}$$

and micro-canonical window width $\Delta = 0.1$. This yields

$$\chi_\star(0, \sigma h_S) = c_1 \tilde{\nu}(0, \sigma h_S)\rho(\sigma h_S) \approx 2^L \times 0.0052. \tag{67}$$

## 3.4 The extent of the weak coupling regime

A given spin-bath Hamiltonian is in the weak coupling regime if the perturbative correction of every eigenstate is small $J^2 \chi_\alpha \ll 1$. Due to the heavy tail of $f_{FS}$ this is a much more stringent condition than requiring the typical eigenstates to be in the perturbative regime. Specifically we find

$$\exp\left[\log \max_\alpha \chi_\alpha\right] \approx d^2 \chi_\star, \tag{68}$$

so that the weak coupling regime corresponds to

$$J\sqrt{\chi_\star} \approx g \ll \frac{1}{d}. \tag{69}$$

# 4 Eigenstate entanglement entropies

We now show how $f_{FS}$ may be used to characterise the statistical properties of the eigenstates in the intermediate and strong coupling regimes. Specifically, we obtain the distribution of entanglement entropies $f_{EE}(S)$ and we numerically verify this claim. This is possible as (i) $\chi_\alpha$ accurately determines the entanglement entropy in both limit of $J^2 \chi_\alpha \ll 1$, where the entropy $S_\alpha$ may be calculated in perturbation theory, and $J^2 \chi_\alpha \gg 1$, where $S_\alpha = \log 2$ (ii) the broad distribution of $\chi_\alpha$ ensures only a negligible fraction of states are in neither of these limits.

Naively $\chi_\alpha$ provides a characterisation of the entanglement entropies $S_\alpha$ only in the perturbative limit, $J\sqrt{\chi_\star} \ll 1/d$, where the series expansion (31) applies. Whilst, at the opposite extreme, typical eigenstates are strongly hybridised by the interaction when typical values of $J^2 \chi_\alpha$ become comparable to unity. This defines the strong coupling regime, $J\sqrt{\chi_\star} \gtrsim 1$, in which the combined system of spin and bath approaches ETH. Between the strong and weak coupling regimes is the intermediate regime

$$\frac{1}{d} \lesssim J\sqrt{\chi_\star} \ll 1, \tag{70}$$

in which the coupling is strong enough to successfully "compete" with the energetic scale of the unperturbed model (specifically the level spacing), but the coupling remains too weak to induce the system to full thermalisation. In this regime a finite fraction of levels are participating in strong "accidental" resonances, with $J^2 \chi_\alpha \gtrsim 1$, despite typical levels satisfying $J^2 \chi_\alpha \ll 1$.

Accidental resonances occur when two neighbouring levels from opposite sectors, $\alpha = (\uparrow, a)$ and $\beta = (\downarrow, b)$, have, by chance, a level separation $\Delta_{\alpha\beta} := \mathcal{E}_\alpha^0 - \mathcal{E}_\beta^0$ which is atypically small $|\Delta_{\alpha\beta}| \ll \rho_0^{-1}$. In such a situation, this two-level resonance dominates the values of the $\chi_\alpha, \chi_\beta$,

thus we approximate by treating them as equal $J^2\chi_\alpha \approx J^2\chi_\beta \approx \left|\mathcal{V}_{\alpha\beta}/\Delta_{\alpha\beta}\right|^2$. These sparse resonances may be treated individually by diagonalising the two level effective Hamiltonian

$$\mathcal{H}_{\text{eff.}} := \begin{pmatrix} \Delta_{\alpha\beta} & \mathcal{V}_{\alpha\beta} \\ \mathcal{V}_{\alpha\beta} & 0 \end{pmatrix} = \Delta_{\alpha\beta} \begin{pmatrix} 1 & J\sqrt{\chi_\alpha} \\ J\sqrt{\chi_\alpha} & 0 \end{pmatrix}. \tag{71}$$

We refer to this approximation scheme as the *two level resonance model*. Within this model, the eigenstates $|\mathcal{E}_\alpha\rangle$ may be exactly calculated

$$|\mathcal{E}_\alpha\rangle = \sqrt{q_\alpha}|\mathcal{E}_\alpha^0\rangle + \sqrt{p_\alpha}|\mathcal{E}_\beta^0\rangle, \tag{72}$$

where we have defined the "transition probability"

$$p_\alpha = p\left(J^2\chi_\alpha\right), \quad q_\alpha = q\left(J^2\chi_\alpha\right), \tag{73}$$

where

$$p(x) := 1 - q(x) := \frac{1}{2}\left(1 - \frac{1}{\sqrt{1+4x}}\right). \tag{74}$$

The exact eigenstates of the Spin-ETH model are not given by (72) due to hybridisation with other states $|\mathcal{E}_\gamma^0\rangle$ at first order and higher order corrections. However, these corrections do not significantly correct the statistics of transition probabilities. We discuss the validity of the two level resonance model in Sec. 8.

Within the two-level resonance model the eigenstate entanglement entropies may be exactly calculated

$$S_\alpha = S(J^2\chi_\alpha), \tag{75}$$

where we have defined

$$S(x) := -p(x)\log p(x) - q(x)\log q(x). \tag{76}$$

To build confidence in this picture of the eigenstates we make some sanity checks. We note that (75) reproduces (31) in the weak coupling limit, and approaches $S_\alpha = \log 2$ for strong hybridisation $J^2\chi_\alpha \gg 1$. As $f_{\text{FS}}$ decays rapidly for $\chi \lesssim \chi_\star$, for $J^2\chi_\star \gg 1$ that all mid-spectrum eigenstates will have $S_\alpha \approx \log 2$ consistent with the spin-bath system approaching ETH in this limit.

For further affirmation we look to numerics. To numerically verify (75) using the Spin-ETH model: (i) we diagonalise the decoupled Hamiltonian $\mathcal{H}_0$ and calculate the fidelity susceptibility $\chi_\alpha$ for each state; (ii) we then diagonalise $\mathcal{H} = \mathcal{H}_0 + \mathcal{V}$ and calculate the von Neumann entropy of the probe spin $S_\alpha$ for each state; and (iii) we identify eigenstates $|\mathcal{E}_\alpha\rangle$ of $\mathcal{H}$ and the eigenstates $|\mathcal{E}_\alpha^0\rangle$ of $\mathcal{H}_0$ by globally maximising the objective function $\prod_\alpha \left|\langle\mathcal{E}_\alpha|\mathcal{E}_\alpha^0\rangle\right|^2$[3]. The pairs $(J^2\chi_\alpha, S_\alpha)$ we obtain are plotted in Fig 5 (coloured points, $J$ values inset), each series of data consists of $N = 200$ mid spectrum states from a single diagonalisation. These points are to be compared with the function $S(J^2\chi_\alpha)$ as given by (76) (black dashed line). As expected the agreement is exact in the limits of large and small $J^2\chi_\alpha$, corresponding to ETH value $S_\alpha = \log 2$ and the perturbative limit respectively. The deviation of $S_\alpha$ from $S(J^2\chi_\alpha)$ is only apparent over a small $O(1)$ region highlighted by the grey box.

The inset in Fig 5 is a density plot of $f(S_\alpha|J^2\chi_\alpha)$, the conditional probability of obtaining a value of the von-Neumann entanglement entropy $S_\alpha$ given a fixed value of $J^2\chi_\alpha$. From the density plot it is apparent that the typical deviation of $S_\alpha$ from $S(J^2\chi_\alpha)$ is significantly smaller than a single decade, and thus, to a reasonable degree of approximation, we may take $S_\alpha$ to be given by $S(J^2\chi_\alpha)$, as in (75). The distribution $f(S_\alpha|J^2\chi_\alpha)$ shown in this plot is calculated using $(J^2\chi_\alpha, S_\alpha)$ aggregated from the mid-spectrum states of $N = 100$ diagonalisations with $\log J$ drawn uniformly and iid from the interval $\log J \in [-10, 2]$.

---

[3]This is a "maximum-weight-matching" problem which can be solved in $O(d^3)$ time by e.g. the Blossom algorithm

## 4.1 Distribution of eigenstate entanglement entropies

Using the distribution of fidelity susceptibility $f_{\mathrm{FS}}(\chi)$, and the two level resonance model for the entanglement entropy $S_\alpha = S(J^2 \chi_\alpha)$, we now calculate the distribution of entanglement entropies

$$f_{\mathrm{EE}}(S|J,E,h_{\mathrm{S}}) = \int \mathrm{d}\chi \, f_{\mathrm{FS}}(\chi|E,h_{\mathrm{S}}) \, \delta(S - S(J^2 \chi)) \tag{77}$$

and show it to agree well with numerical calculations of $f_{\mathrm{EE}}$. We analyse this distribution highlighting two quantitative features. The first is a simple universal form at entropies above

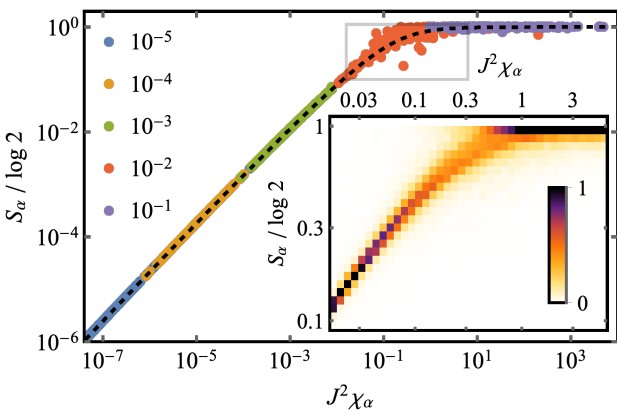

Figure 5: *Values of* $(S_\alpha, \chi_\alpha)$: The entanglement entropy of $S_\alpha$ of a eigenstate of the Spin-ETH model as a function of the fidelity susceptibility of the corresponding eigenstates of $\mathcal{H}_0$ ($J = 0$). For each value of the coupling $J$ (coloured points, $J$ values in legend), $N = 200$ points corresponding to randomly selected mid-spectrum eigenstates are shown. Inset a histogram of data aggregated across many diagonalisations showing the distribution within the grey boxed region of the main plot. Each column of cells in the inset is normalised to sum to unity. Parameters $L = 12$, other parameters as in Fig. 4.

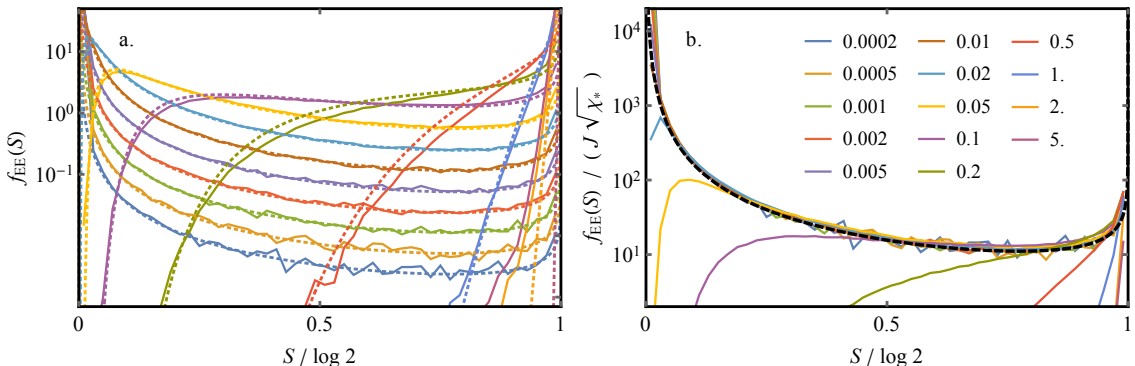

Figure 6: *Distribution of entanglement entropies*: The distribution of spin entanglement entropies in the $L = 12$ Spin-ETH model is numerically extracted for coupling strengths (values of $J\sqrt{\chi_\star(0,h_{\mathrm{S}})}$ inset in right panel). a) data for each coupling strength is plotted (solid colours) together with the predicted analytic form (78) (dashed line). b) data from the left panel is collapsed in accordance with (80), and plotted with the theoretical curve (black dashed line). $N = 800$ realisations per data series.

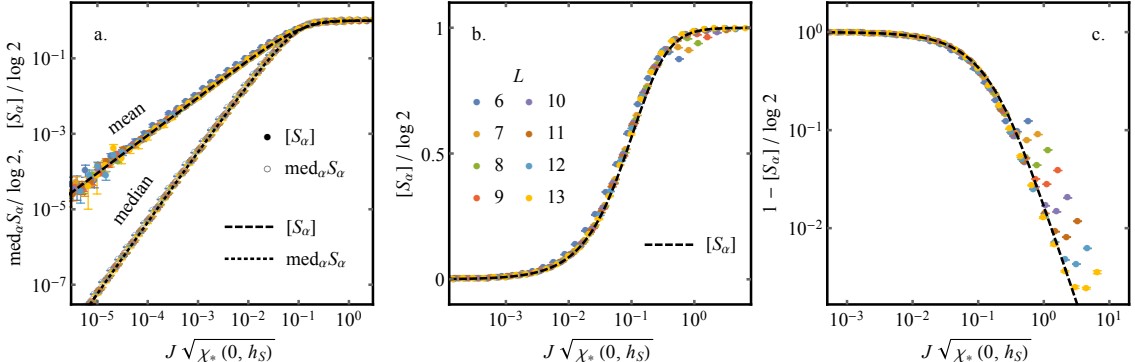

Figure 7: *Mean and median eigenstate entanglement entropy of the spin as a function of coupling strength*: three panels of the same data which plot the analytic form (81a) (no fit parameters) for the mean entanglement entropy (black dashed) with numerical data (coloured solid points). The panels (a), (b) and (c) are plotted to emphasise the lower tail, crossover region, and upper tail respectively (note the change in y-axis for (c)). As the median is asymptotically separated from the mean in the lower tail we additionally plot this quantity in the left panel. The analytic form for the median (82a) is shown (black dotted) together with numerical data (coloured hollow points). $N = 2000$ realisations per data point for $L \leq 10$, $N = 300$ for $L = 11, 12$, $N = 10$ for $L = 13$.

the typical value $S \gg S(J^2 \chi_\star)$. The second is a separation of mean and typical entanglement entropies, which is due rare resonances dominating the mean.

### 4.1.1 Universal form for $f_{\text{EE}}$

We extract the distribution of entanglement entropies by performing the integral (77)

$$f_{\text{EE}}(S|J, E, h_{\text{S}}) = f_{\text{FS}}\left(\frac{x(S)}{J^2}\bigg|E, h_{\text{S}}\right)\frac{1}{J^2 S'(x(S))}. \tag{78}$$

The typical entanglement entropy $S \gg S(J^2 \chi_\star)$, the distribution of fidelity susceptibilities is well approximated by its limiting form

$$f_{\text{FS}} = \sqrt{\frac{\chi_\star}{\chi^3}} + O\left(\frac{\chi_\star}{\chi^2}\right), \tag{79}$$

yielding a correspondingly simplified distribution of entanglement entropies

$$f_{\text{EE}}(S|J, E, h_{\text{S}}) = \frac{J\sqrt{\chi_\star}}{x(S)^{3/2}S'(x(S))} + O(J^2\chi_\star). \tag{80}$$

We comment on the shape of the distribution $f_{\text{EE}}$. The bi-modality of the distribution follows from the compression of the long tail of $f_{\text{FS}}$ onto the bounded interval $S_\alpha \in [0, \log 2]$, producing a second mode at maximal entropy $S = \log 2$. This is in addition to the dominant mode at $S \approx 0$, which contains the median, and corresponds to the single mode of $f_{\text{FS}}$. Secondly we note that (80) implies a scaling collapse of $f_{\text{EE}}(S|J, E, h_{\text{S}})$ upon dividing by $J\sqrt{\chi_\star}$.

In Fig. 6 we numerically verify (78) and (80). We plot the distribution $f_{\text{EE}}$ of spin eigenstate entanglement entropies in the Spin-ETH model for bath size $L = 12$. In Fig. 6a a histogram of numerically calculated $S_\alpha$ values is plotted (solid lines) for mid-spectrum states for various values of $J\sqrt{\chi_\star}$. The values of, $J\sqrt{\chi_\star(0, h_{\text{S}})}$ (inset, right panel) are calculated using (67). These

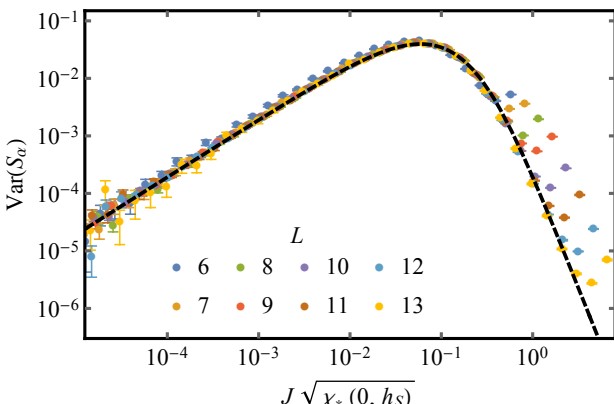

Figure 8: *Variance of the eigenstate entanglement entropies of the spin*: the numerically ensemble averaged variance (coloured points) is plotted as a function of $J\sqrt{\chi(0,h_S)}$ for mid-spectrum states of the Spin-ETH model for different $L$ (legend inset). Data corresponds to the same realisations as Fig. 7. The theoretical curve (black dashed) is calculated from the distribution (78).

numerical estimates of $f_{EE}$ are compared with the analytic form (78) (dotted lines) calculated using $f_{FS}$ as in (51b). The predicted and measured curves agree exactly in the intermediate coupling regime ($J\sqrt{\chi_\star(0,h_S)} \ll 1$), whereas there is some discrepancy associated with crossover into the strong coupling regime ($J\sqrt{\chi_\star(0,h_S)} \gtrsim 1$) due to the inexact nature of the two-level resonance model. In Fig. 6b we show the predicted scaling collapse by plotting the same data but vertically re-scaled by $J\sqrt{\chi_\star(0,h_S)}$. The re-scaled data collapses onto the form predicted by (80) (black, dashed line) for entropies above the typical value $S \gg S(J\sqrt{\chi_\star})$. As the typical value becomes comparable to $S = \log 2$ the lower mode disappears, and $f_{EE}$ has a single mode close to the thermal entropy $S = \log 2$.

In Figs. 7 and 8 we compare the analytic and numerical calculations of the mean, median and variance of the entanglement entropy (using the same diagonalisations as Fig 6). The three panels of Fig. 7 show the same $[S_\alpha]$ data plotted to emphasise the agreement at small, intermediate and large values of $J\sqrt{\chi_\star}$ respectively. Good agreement is found between the analytic (dashed lines) and numerically calculated values of $[S_\alpha]$ (solid colour points) across all values of $J\sqrt{\chi_\star}$. There is deviation at large $J\sqrt{\chi_\star}$ (Fig 7), where the numerical data peels off from the theoretical curve. The magnitude of this deviation decreases exponentially decreasing with $L$.

### 4.1.2 Limit of weak coupling $J^2\chi_\star \ll 1$: separation of mean and typical behaviour

We now extract the analytical form of the limiting behaviours of the mean $[S_\alpha]$, median $\mathrm{med}_\alpha S_\alpha$, and variance $\mathrm{Var}(S_\alpha)$ of the entanglement entropies within the two level resonance model.

We first consider the mean entanglement entropy in the weak coupling limit. In the limit of small $J\sqrt{\chi_\star}$ we may replace $f_{FS}$ with its large $\chi$ asymptotic form (79) and expand in powers of $J\sqrt{\chi_\star}$

$$[S_\alpha] = \int \mathrm{d}\chi\, f_{FS}(\chi|E,h_S)S(J^2\chi) \tag{81a}$$

$$= 2\pi J\sqrt{\chi_\star} + O(J^2\chi_\star). \tag{81b}$$

The behaviour of the mean in the weak coupling limit may be contrasted by the asymptotically

faster decay of the median

$$\text{med}_\alpha S_\alpha = S(J^2 \, \text{med}_\alpha \chi_\alpha) \tag{82a}$$

$$= (1 - \log c_{\text{m.}} J^2 \chi_\star) c_{\text{m.}} J^2 \chi_\star + O(J^4 \chi_\star^2), \tag{82b}$$

where $c_{\text{m.}} := \text{med}_\alpha \chi_\alpha / \chi_\star$ is some $O(1)$ constant. The asymptotic separation of the mean (solid coloured circles) and median (hollow coloured circles) is visible in Fig. 7a.

We may also obtain the variance from the same approach. First we calculate the second moment of the entanglement entropy

$$[S_\alpha^2] = \int \mathrm{d}\chi \, f_{\text{FS}}(\chi | E, \omega) S^2(J^2 \chi) = c_{\text{v.}} J \sqrt{\chi_\star} + O(J^2 \chi_\star), \tag{83}$$

where $c_{\text{v.}} = \int_0^\infty \mathrm{d}x S(x) x^{-3/2} = 1.91755\ldots$. This yields a variance

$$\text{Var}(S_\alpha) = [S_\alpha^2] - [S_\alpha]^2 = c_{\text{v.}} J \sqrt{\chi_\star} + O(J^2 \chi_\star). \tag{84}$$

### 4.1.3 Limit of strong coupling $J^2 \chi_\star \gg 1$

The distribution $f_{\text{FS}}$ decays rapidly for $\chi \lesssim \chi_\star$, as such we may expand $S(x) = \log 2 - 1/(8x) + O(x^{-2})$ yielding

$$[S_\alpha] = \log 2 - \frac{c_{\text{a.}}}{8 J^2 \chi_\star} + O(J^2 \chi_\star)^{-2}, \tag{85}$$

where $c_{\text{a.}} = \chi_\star [\chi_\alpha^{-1}]$ is an $O(1)$ numerical constant. Following the same approach for the variance yields

$$\text{Var}(S_\alpha) = \frac{c_{\text{v.}}'}{64 J^4 \chi_\star^2} + O(J^2 \chi_\star)^{-3}, \tag{86}$$

where $c_{\text{v.}}' = \chi_\star^2 \text{Var}(\chi^{-1})$ is again an $O(1)$ constant.

We note that (85) agrees with ETH; in contrast $\text{Var}(S_\alpha)$ is smaller than the ETH prediction of $\text{Var}(S_\alpha) \propto (J^2 \chi_\star)^{-1}$. This discrepancy follows from the $O(1/d)$ scale of the off-diagonal elements of the density matrix (30), which holds in the limit of fixed $J \sqrt{\chi_\star}$ as $d \to \infty$. In this limit we do not recover the off-diagonal ETH for spin observables. The fluctuations predicted by ETH are obtained only in the regime $J/h_S$ held fixed as $d \to \infty$, when the off-diagonal elements of the density matrix $\hat\rho_\alpha$ are $O(1/\sqrt{d})$.

## 5 Infinite time memory in dynamical evolution

The Spin-ETH model consists of a few level system weakly coupled to a thermal bath, and is thus a prototypical setting for applying Fermi's Golden Rule (FGR), which predicts the exponential decay two-time correlators. However, in the weak and intermediate coupling regime, the spin maintains appreciable memory of its initial conditions even at infinite time, a feature not captured by FGR. We show that the two-level resonance model provides a quantitative description of this infinite time memory.

### 5.1 Strong coupling limit $J \sqrt{\chi_\star} \gg 1$

Let us recall the predictions of FGR. Consider a system prepared in an eigenstate $|\mathcal{E}_\alpha^0\rangle = |\uparrow\rangle |E_a\rangle$ of the decoupled Hamiltonian $\mathcal{H}_0$. Dynamical evolution under the full Hamiltonian $\mathcal{H}$ will cause population to leak from $|\mathcal{E}_\alpha^0\rangle$ into a set of target states $|\mathcal{E}_\beta^0\rangle = |\downarrow\rangle |E_b\rangle$ at the target

energy $E_b \approx E_a + h_S$ (and subsequently on-wards into states $|\mathcal{E}_\gamma^0\rangle = |\uparrow\rangle|E_c\rangle$). FGR states that the rate $\Gamma_\alpha$ of population leakage out of the state $|\mathcal{E}_\alpha^0\rangle$ is set by the size of the typical matrix element, and the density of states at the target energy

$$
\begin{aligned}
\Gamma_\uparrow(E_a) &= 2\pi |J\, V_{ab}|^2 \rho(E_a + h_S) \\
&= 2\pi J^2 \tilde{v}(E_a, h_S),
\end{aligned} \tag{87}
$$

using ETH ansatz (57).

The decay of the initial state populations causes a decay in two-time correlations. For specificity we consider the connected $zz$ correlator evaluated with an initial infinite-temperature state

$$
C_{zz}(t) := \frac{1}{2d} \operatorname{tr}\left( e^{i\mathcal{H}t} (\sigma^z \otimes \mathbb{1}) e^{-i\mathcal{H}t} (\sigma^z \otimes \mathbb{1}) \right). \tag{88}
$$

The FGR does not account for the finite nature of the bath, and thus predicts indefinite exponential decay of correlations

$$
\log C_{zz}(t) = -\gamma t + O(t^2/L), \tag{89}
$$

with an exponential decay rate (derived in Appendix D)

$$
\gamma = \frac{2\pi J^2}{d} \sum_\sigma \int dE \rho(E) \tilde{v}(E, \sigma h_S). \tag{90}
$$

For the Spin-ETH model studied in the manuscript, evaluating (90) numerically yields $\gamma/J^2 = 1.64\ldots$.

## 5.2 Intermediate and weak coupling $J\sqrt{\chi_\star} \ll 1$

In contrast to the indefinite exponential decay predicted by the FGR, in the the weak and intermediate coupling regime $J\sqrt{\chi_\star} \lesssim 1$ many eigenstates of the system are only weakly entangled. These cause the spin to maintain appreciable memory of its initial state at infinite time. This infinite memory can be quantified in the infinite time average of the spin-spin correlator

$$
\overline{C}_{zz} = \lim_{t \to \infty} \frac{1}{t} \int_0^t dt'\, C_{zz}(t') = \frac{1}{2d} \sum_\alpha \langle \mathcal{E}_\alpha | \sigma^z | \mathcal{E}_\alpha \rangle^2, \tag{91}
$$

where the second equality is obtained by expanding in the eigenbasis. Only eigenstates which are close to product states contribute to $\overline{C}_{zz}$, which is thus approximately proportional to the fraction of eigenstates in the lower mode of $f_{EE}$. More precisely, we may evaluate (91) within the two level resonance model

$$
\langle \mathcal{E}_\alpha | \sigma^z | \mathcal{E}_\alpha \rangle^2 = \left(1 - 2p(J^2 \chi_\alpha)\right)^2 = \frac{1}{1 + 4J^2 \chi_\alpha}. \tag{92}
$$

We obtain an analytic form for the infinite time correlator by first ensemble averaging

$$
[\langle \mathcal{E}_\alpha | \sigma^z | \mathcal{E}_\alpha \rangle^2] = \int d\chi f(\chi | E_a, \sigma h_S) \frac{1}{1 + 4J^2 \chi}, \tag{93}
$$

(where $\alpha = (a, \sigma)$), and subsequently summing over possible states $\alpha$ to obtain

$$
\overline{C}_{zz} = \frac{1}{2d} \sum_{\sigma \in \uparrow, \downarrow} \int dE\, \rho(E) \int d\chi\, f_{FS}(\chi | E, \sigma h_S) \frac{1}{1 + 4J^2 \chi}. \tag{94}
$$

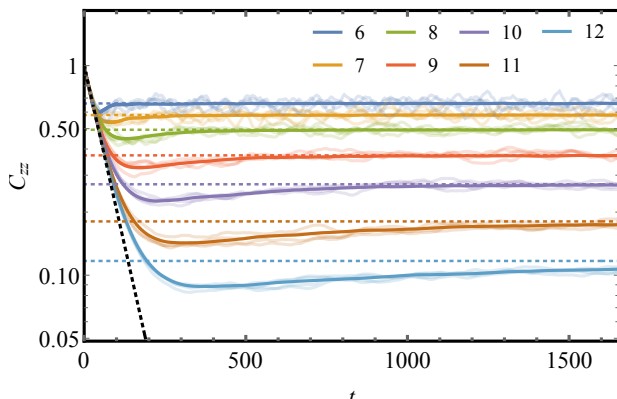

Figure 9: *Finite time correlations* in the Spin-ETH model for coupling strength $J = 0.1$ and different system sizes $L$ (legend). The correlator $C_{zz}(t)$ initially decays exponentially with FGR setting the decay rate (black dashed line). For a finite bath, the ensemble averaged correlations (coloured solid lines) saturate to a finite value which we extract numerically (coloured dashed lines). Individual trajectories (coloured translucent lines) exhibit small oscillations around this value. The numerically extracted saturation values are compared with theoretical values in Fig 10.

The weak coupling behaviour of $\overline{C}_{zz}$ is given by

$$\overline{C}_{zz} = 1 - 4\pi J \sqrt{\frac{\chi_\star(0, h_\mathrm{S})}{6}} + O(J^2 \chi_\star(0, h_\mathrm{S})) + O(L^{-1/2}). \tag{95}$$

To recover (95) we consider the following quantity $K$ which must be shown to have value $K = 4\pi/\sqrt{6}$:

$$
\begin{aligned}
K &= \lim_{J \to 0} \frac{1 - \overline{C}_{zz}}{J\sqrt{\chi_\star(0, h_\mathrm{S})}} \\
&= \lim_{J \to 0} \frac{1}{2d} \int \mathrm{d}E \, \rho(E) \int \mathrm{d}\chi \sum_{\sigma \in \uparrow, \downarrow} \frac{f_\mathrm{FS}(\chi | E, \sigma h_\mathrm{S})}{\sqrt{\chi_\star(0, h_\mathrm{S})}} \frac{4J\chi}{1 + 4J^2\chi} \\
&= \frac{\pi}{d} \int \mathrm{d}E \, \rho(E) \sum_{\sigma \in \uparrow, \downarrow} \sqrt{\frac{\chi_\star(E, \sigma h_\mathrm{S})}{\chi_\star(0, h_\mathrm{S})}} \\
&= \frac{2\pi}{d} \int \mathrm{d}E \, \rho(E) \sqrt{\frac{\rho(E)}{\rho(0)}} + O(L^{-1/2}).
\end{aligned}
\tag{96}
$$

Here, in the second line we have substituted (94), and in the third line we have used that $f_\mathrm{FS} \sim \chi_\star^{1/2}/\chi^{3/2}$ at large $\chi$, and performed the resulting integral $\int \mathrm{d}x \, x^{-3/2} 4x/(1+4x) = 2\pi$. To obtain the final line we have then used

$$
\begin{aligned}
\chi_\star(E, \sigma h_\mathrm{S}) &= c_\beta \tilde{v}(E, \sigma h_\mathrm{S}) \rho(E + h_\mathrm{S}) \\
&= c_\beta \tilde{v}(0, h_\mathrm{S}) \rho(E) + O(L^{-1/2}) \\
&= \chi_\star(0, h_\mathrm{S}) \rho(E)/\rho(0) + O(L^{-1/2}).
\end{aligned}
\tag{97}
$$

Performing the Gaussian integral in the final line of (96) we obtain the desired result $K = 4\pi/\sqrt{6}$, and hence (95) follows.

In Figs 9 and 10 we numerically verify the saturation values $\overline{C}_{zz}$ of the two-time spin correlator, (94) and (95) in the Spin-ETH model. In Fig 9, for bath of size $L$ (legend inset) we show a sub-sample of $N = 4$ trajectories (translucent colours) and the sample mean value of $C_{zz}(t)$

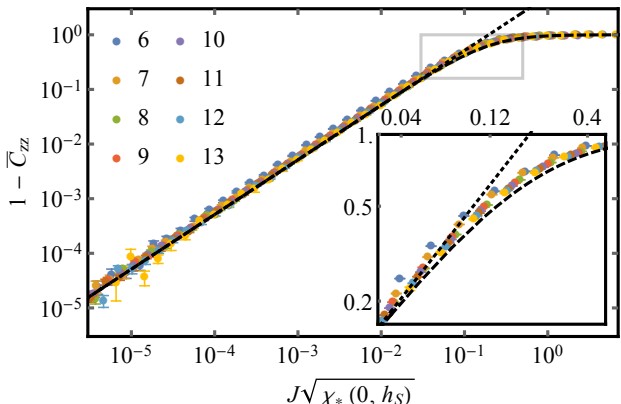

Figure 10: *Infinite time correlations*: The infinite time spin correlations $\overline{C}_{zz}$ are plotted for the Spin-ETH model. The predicted theoretical form (black dashed) crosses over from $\overline{C}_{zz} \to 1$ as $J \to 0$ to $\overline{C}_{zz} \to 0$ as $J\sqrt{\chi_\star(0, hp)} \gg 1$. The small $J$ asymptote (95) is also shown (black dotted). The theoretical forms show good agreement with numerically extracted values (coloured solid lines, $L$ values on legend, inset). The region enclosed within the grey box where $\overline{C}_{zz}$ crosses over between its limiting values is shown (plot inset). $N = 1000, 100, 10$ realisations per data point for $L = 11, 12, 13$ repsectively and $N = 3000$ for $L \leq 10$.

(solid colours),. These trajectories track the FGR prediction (89) at early times (black dashed) before converging to the ensemble averaged infinite time value (dashed colour). The convergence from below is related to the well known 'dip' and 'ramp' features of the spectral form factor in Gaussian random matrices systems [35]. In Fig. 10 the numerically calculated infinite time saturation values $\overline{C}_{zz}$ (solid colours) are compared with theoretical predictions (94) (black dashed). The agreement is good throughout the plot range. The weak coupling approximation (95) (black dotted), also shows good agreement for $J\sqrt{\chi_\star} \ll 1$.

# 6 Off-diagonal matrix elements of operators on the bath

In the weak and intermediate regimes, the spin-bath system does not satisfy ETH. However, operators on the bath do satisfy an ETH-like ansatz in which off-diagonal matrix elements within a small spectral window have a non-Gaussian distribution. This distribution deforms smoothly between the weak coupling limit ($J\sqrt{\chi_\star} \ll 1/d$), wherein ETH is satisfied on the bath (but not the combined spin-bath system), and the strongly coupled limit ($J\sqrt{\chi_\star} \gtrsim 1$) where the entire spin-bath system approaches ETH.

Consider the weak coupling regime. A local operator $V$ on the bath satisfies ETH (57) with the random numbers $R_{ab}$ being Gaussian distributed [54–59]. Two arguments help see why the $R_{ab}$ are Gaussian distributed in ETH: (i) the distribution of the $R_{ab}$ is constrained only by $[R_{ab}] = 0$ and $[R_{ab}^2] = 1$, and the standard normal distribution is the maximum entropy distribution with this property (i.e. deviation from normality would imply the existence of additional constraints) and (ii) under fairly weak assumptions (violated in the case of e.g. localisation), Gaussian distributed elements represent the only perturbatively stable situation. To see this consider a weak perturbation to the bath $H_B \to H_B' = H_B + \Delta H$. Let the energy scale $|\Delta H|$ of this perturbation be much larger than the level spacing, but much smaller than the local bandwidth so that only states for which $\bar{V}(E)$ and $\tilde{v}(E, \omega)$ have essentially the same value hybridise. Consider the matrix elements of $V$ in the new eigenbasis: the functions $\bar{V}(E)$

and $\tilde{v}(E, \omega)$ are unaltered from (57), but the $R_{ab}$ coefficients linearly superpose:

$$R_{ab} \rightarrow R'_{ab} = \sum_{cd} U_{ac} R_{cd} U^{\dagger}_{db}. \tag{98}$$

Above, $U$, the unitary which maps from the unperturbed to the perturbed eigenbasis, superposes unperturbed levels with small energy separations $|E_a - E_b| \lesssim |\Delta H|$. As $R'_{ab}$ is a weighted sum of the $R_{ab}$, by the central limit theorem, it is normally distributed.

At zero coupling the bath satisfies ETH. However, the combined spin-bath system does not, as the off-diagonal matrix are not Gaussian distributed. The matrix elements of $\mathbb{1} \otimes V$ evaluated between eigenstates $\alpha = (a, \sigma)$ and $\beta = (b, \tau)$ of $\mathcal{H}$ are given by

$$
\begin{aligned}
V_{\alpha\beta} &:= \langle \mathcal{E}_{\alpha} | \mathbb{1} \otimes V | \mathcal{E}_{\beta} \rangle \\
&= \bar{V}(\mathcal{E}_{\alpha}) \delta_{\alpha\beta} + \sqrt{\frac{\tilde{v}(\mathcal{E}_{\alpha}, \mathcal{E}_{\beta} - \mathcal{E}_{\alpha})}{2\rho(\mathcal{E}_{\beta})}} \, \mathcal{R}_{\alpha\beta}.
\end{aligned}
\tag{99}
$$

Above $2\rho(E)$ is the density of states of the combined spin-bath system. In (99), and throughout this section, we neglect the $O(L^{-1})$ correction to the energy density of the system from the spin so that $\bar{V}(E_a) = \bar{V}(\mathcal{E}_{\alpha}) + O(L^{-1})$. In decoupled limit $J \rightarrow 0$, the random matrix elements are given by

$$\mathcal{R}_{\alpha\beta} = \sqrt{2} \delta_{\sigma\tau} R_{ab}. \tag{100}$$

The $\mathcal{R}_{\alpha\beta}$ are strongly non Gaussian: half the elements $\mathcal{R}_{\alpha\beta}$ are exactly zero, whereas half are Gaussian distributed with twice the variance predicted by ETH. In the strong coupling regime the $\mathcal{R}_{\alpha\beta}$ will be Gaussian distributed with $[\mathcal{R}_{\alpha\beta}] = 0$, $[|\mathcal{R}_{\alpha\beta}|^2] = 1$ as required.

We characterise the crossover between the strong and weak coupling regimes by evaluating the distribution of off-diagonal matrix elements on the bath within the two-level resonance model. Consider the matrix element $V_{\alpha\beta}$ between the two eigenvectors of the first spin and bath

$$
\begin{aligned}
|\mathcal{E}_{\alpha}\rangle &= \sqrt{q_{\alpha}} |\mathcal{E}_{\alpha}^0\rangle + \sqrt{p_{\alpha}} |\mathcal{E}_{\gamma}^0\rangle, \\
|\mathcal{E}_{\beta}\rangle &= \sqrt{q_{\beta}} |\mathcal{E}_{\beta}^0\rangle + \sqrt{p_{\beta}} |\mathcal{E}_{\delta}^0\rangle.
\end{aligned}
\tag{101}
$$

Here the $|\mathcal{E}^0\rangle$ are product states of the spin and bath, with the subscripts $\alpha = (\sigma, a)$, $\beta = (\tau, b)$, $\gamma = (-\sigma, c)$ and $\delta = (-\tau, d)$.

There are two distinct cases of off-diagonal elements to consider: the even case $\sigma = \tau$, and the odd case $\sigma = -\tau$. Taking the even case first, we use the ETH ansatz (57) to obtain the matrix element

$$
\begin{aligned}
V_{\alpha\beta}^{(e)} &= \langle \mathcal{E}_{\alpha} | \mathbb{1} \otimes V | \mathcal{E}_{\beta} \rangle \\
&= \sqrt{q_{\alpha} q_{\beta}} \langle \mathcal{E}_{\alpha}^0 | \mathbb{1} \otimes V | \mathcal{E}_{\beta}^0 \rangle + \sqrt{p_{\alpha} p_{\beta}} \langle \mathcal{E}_{\gamma}^0 | \mathbb{1} \otimes V | \mathcal{E}_{\delta}^0 \rangle \\
&= \sqrt{\frac{\tilde{v}(\mathcal{E}_{\beta} - \mathcal{E}_{\alpha})}{2\rho_0}} \mathcal{R}_{\alpha\beta}^{(e)},
\end{aligned}
\tag{102}
$$

where the random coefficient

$$\mathcal{R}_{\alpha\beta}^{(e)} := R_{ab} \sqrt{2 q_{\alpha} q_{\beta}} + R_{cd} \sqrt{2 p_{\alpha} p_{\beta}} \tag{103}$$

has mean and variance

$$[\mathcal{R}_{\alpha\beta}^{(e)}] = 0, \quad [|\mathcal{R}_{\alpha\beta}^{(e)}|^2] = 2[q_{\alpha} q_{\beta} + p_{\alpha} p_{\beta}]. \tag{104}$$

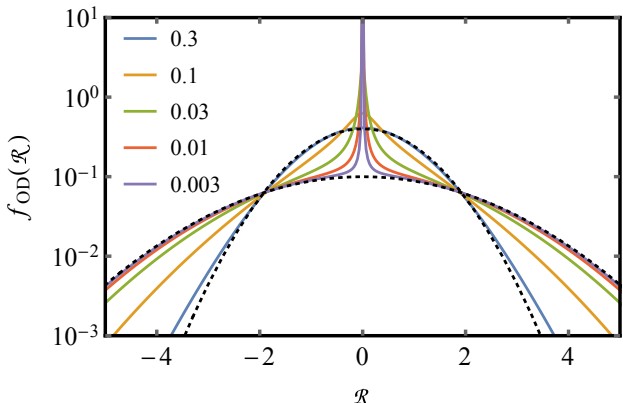

Figure 11: *Distribution $f_{\mathrm{OD}}(\mathcal{R})$ of the off-diagonal matrix elements of operators on the bath*: Eq. (107) is plotted for different values of $J\sqrt{\chi_\star}$ from the intermediate regime (values in legend, inset). The dotted lines show the limiting cases of weak coupling, where $f_{\mathrm{OD}}(\mathcal{R}) \to \frac{1}{2}f_{\mathrm{N}}(\mathcal{R}|0,2) + \frac{1}{2}\delta(\mathcal{R})$, and strong coupling $f_{\mathrm{OD}}(\mathcal{R}) \to f_{\mathrm{N}}(\mathcal{R}|0,1)$ (where $f_{\mathrm{N}}(\mathcal{R}|\mu,\sigma^2)$ is the normal distribution).

We now obtain the distribution for $\mathcal{R}_{\alpha\beta}^{(\mathrm{e})}$. Let $f_{\mathrm{N}}(R|\mu,\sigma^2)$ denote the usual normal distribution of mean $\mu$ and variance $\sigma^2$. The $R_{ab}$ are distributed as $f_{\mathrm{N}}(R|0,1)$, while $p_\alpha = p(J^2\chi_\alpha)$ with $\chi_\alpha$ distributed according to $f_{\mathrm{FS}}(\chi)$. Thus,

$$f_{\mathrm{OD}}^{(\mathrm{e})}(\mathcal{R}) = \iint d\chi\, d\chi'\, f_{\mathrm{FS}}(\chi) f_{\mathrm{FS}}(\chi') f_{\mathrm{N}}\big(\mathcal{R}|0, v^{(\mathrm{e})}(J^2\chi, J^2\chi')\big), \qquad (105)$$

where for brevity we have defined $v^{(\mathrm{e})}(x,y) = 2q(x)q(y) + 2p(x)p(y)$. It is readily verified that this distribution has the mean and variance in (104).

Repeating this calculation for the odd case, we obtain $\mathcal{R}_{\alpha\beta}^{(\mathrm{o})}$ with mean and variance

$$[\mathcal{R}_{\alpha\beta}^{(\mathrm{o})}] = 0, \quad [|\mathcal{R}_{\alpha\beta}^{(\mathrm{o})}|^2] = 2[q_\alpha p_\beta + p_\alpha q_\beta], \qquad (106)$$

and corresponding distribution

$$f_{\mathrm{OD}}^{(\mathrm{o})}(\mathcal{R}) = \iint d\chi\, d\chi'\, f_{\mathrm{FS}}(\chi) f_{\mathrm{FS}}(\chi') f_{\mathrm{N}}\big(\mathcal{R}|0, v^{(\mathrm{o})}(J^2\chi, J^2\chi')\big), \qquad (107)$$

with $v^{(\mathrm{o})}(x,y) = 2q(x)p(y) + 2p(x)q(y)$.

In sum, the distribution of off-diagonal elements $\mathcal{R}_{\alpha\beta}$ is given by,

$$f_{\mathrm{OD}}(\mathcal{R}) = \frac{1}{2}\left(f_{\mathrm{OD}}^{(\mathrm{e})}(\mathcal{R}) + f_{\mathrm{OD}}^{(\mathrm{o})}(\mathcal{R})\right). \qquad (108)$$

The distribution $f_{\mathrm{OD}}$ is plotted for different values of $J\sqrt{\chi_\star}$ in Fig. 11. As $J$ is tuned through the intermediate regime $f_{\mathrm{OD}}$ interpolates smoothly between the weak coupling limit of $f_{\mathrm{OD}}(\mathcal{R}) \to \frac{1}{2}f_{\mathrm{N}}(\mathcal{R}|0,2) + \frac{1}{2}\delta(\mathcal{R})$ where ETH is satisfied within each spin sector, and the strong coupling limit of $f_{\mathrm{OD}}(\mathcal{R}) \to f_{\mathrm{N}}(\mathcal{R}|0,1)$ where the combined system approaches ETH. At intermediate values, $f_{\mathrm{OD}}$ is visibly non-Gaussian.

## 7 The entropy of the bath

In the intermediate regime, the *effective* density of states of the bath is enhanced by the partial thermalisation of the spin, $\rho(E) \leq \rho_{\mathrm{eff}} \leq 2\rho(E)$. We characterise this smooth enhancement

with the *matrix element entropy* $\Delta \mathcal{S} = \log(\rho_{\mathrm{eff}}/\rho(E))$ which describes the effective entropy of the bath as felt by a second, weakly coupled, probe spin.

Introduce a second 'probe' spin with field $h'_{\mathrm{S}}$ coupled to the bath in the same manner as the first, with a weak coupling constant $J'$ and bath operator $V'$ (here and throughout this section primed variables relate to the second spin). This second spin sees an "effective bath" composed of $H_{\mathrm{B}}$ together with the first spin, see Fig. 2a. Applying the results of Secs. 3.2 and 3.3, the hybridisation of the states at energy $\mathcal{E}$ is quantitatively characterised by the scalar quantity $J'^2 \chi'_\star(J, \mathcal{E}, h'_{\mathrm{S}})$ where

$$\chi'_\star(J, \mathcal{E}, \omega) := [|V'_{\alpha\beta}|]^2 \rho'(\mathcal{E} + \omega)^2. \tag{109}$$

Here $\rho'(\mathcal{E}) = 2\rho(\mathcal{E}) + O(L^{-1})$ is the density of states of the combined (first) spin+bath. We may also use (64) to define an effective density of states

$$\chi'_\star(J, E, \omega) = c_\beta \, \tilde{v}'(J, E, \omega) \rho_{\mathrm{eff}}(E + \omega), \tag{110}$$

where $\tilde{v}'$ is the spectral function of $V'$.

At weak coupling, $\chi'_\star(0, \mathcal{E}, \omega)$ is given by (44). At strong coupling to the first spin, the typical fidelity susceptibility is twice its $J = 0$ value $\chi'_\star(J, \mathcal{E}, \omega) = 2\chi'_\star(0, \mathcal{E}, \omega)$. Recalling (64), we understand the factor two growth of $\chi'_\star$ as an enhancement of the effective bath density of states $\rho_{\mathrm{eff}}$ due to strong hyrbidisation with the first spin, or equivalently as a $\log 2$ enhancement of bath entropy $\mathcal{S} = \log \rho_{\mathrm{eff}}$ [25, 60, 61]. It is thus natural to define the entropic enhancement of the bath at intermediate values by the *matrix element entropy*

$$\Delta \mathcal{S}(J, \mathcal{E}, h'_{\mathrm{S}}) := \log\left( \frac{\chi'_\star(J, \mathcal{E}, h'_{\mathrm{S}})}{\chi'_\star(0, \mathcal{E}, h'_{\mathrm{S}})} \right) \tag{111a}$$

$$= 2\log\left( \frac{[|V'_{\alpha\beta}|]}{[|V'_{\alpha\beta}|]_{J=0}} \right). \tag{111b}$$

As before, $[|V'_{\alpha\beta}|]$ is the mean absolute value of the matrix elements averaged over levels $\alpha$ and $\beta$ taken from small windows about the energies $\mathcal{E}$ and $\mathcal{E} + h_{\mathrm{S}}$ respectively. $[|V'_{\alpha\beta}|]_{J=0}$ is the same quantity evaluated for zero coupling to the first spin $J = 0$.

We recast the matrix element entropy $\Delta \mathcal{S}$ in terms of more familiar objects: it is the Renyi entropy of order $n = 1/2$ associated to the $\mathcal{R}_{\alpha\beta}$. Specifically, as the $\mathcal{R}_{\alpha\beta}$ square to one $[|\mathcal{R}_{\alpha\beta}|^2] = 1$, we may define the normalised "probability distribution" $\mathcal{P}_{\alpha\beta} = |\mathcal{R}_{\alpha\beta}|^2/\mathcal{N}$ where $\mathcal{N}$ is a normalisation constant, and $\alpha$, $\beta$ are restricted to the aforementioned energy windows. The Renyi entropy of order $n$ associated to this distribution is

$$\mathsf{H}_n(\mathcal{P}) = \frac{1}{1-n} \log\left( \sum_{\alpha\beta} \mathcal{P}_{\alpha\beta}^n \right). \tag{112}$$

Comparing (112), (111a) and (99) we see that

$$\Delta \mathcal{S}(J, \mathcal{E}, h_{\mathrm{S}}) = \mathsf{H}_{1/2}(\mathcal{P}) - \mathsf{H}_{1/2}(\mathcal{P})\big|_{J=0}. \tag{113}$$

We now evaluate the matrix element entropy. Starting from (113) with $\mathcal{P}_{\alpha\beta} = |\mathcal{R}_{\alpha\beta}|^2/\mathcal{N}$ we may perform the $\mathcal{R}$-average using distribution of off-diagonal matrix elements (107) to obtain

$$\Delta \mathcal{S} = 2\log\left( \iint \mathrm{d}\chi \, \mathrm{d}\chi' f_{\mathrm{FS}}(\chi) f_{\mathrm{FS}}(\chi') k(J^2\chi, J^2\chi') \right), \tag{114}$$

where, for brevity, we have suppressed the dependencies of $\Delta S$ and $f_{FS}$, and defined the kernel

$$k(x,y) := \sqrt{p(x)p(y) + q(x)q(y)} + \sqrt{p(x)q(y) + q(x)p(y)}. \tag{115}$$

Eq. (114) is exact within the two level resonance model, but cannot be straightforwardly simplified to a closed form expression. However, in the asymptotic limits of weak and strong coupling simpler forms may be extracted (see Appendix E), yielding respectively

$$\Delta S = -8J\sqrt{\chi_\star} \log(J\sqrt{\chi_\star}) + O(J\sqrt{\chi_\star}), \tag{116a}$$

$$\Delta S = \log 2 + O\big((J^2\chi_\star)^{-2}\big). \tag{116b}$$

The matrix element entropy $\Delta S$ calculated here determines $\chi'_\star$, which in turn sets the large $\chi'$ tail of the distribution of the fidelity susceptibilities $\chi'_{(\alpha,\tau)}$ of the product states $|\tau\rangle|\mathcal{E}_\alpha\rangle$ to switching on the coupling $J'$. $\chi'_{(\alpha,\tau)}$ is defined in precise analogue to (28)

$$\chi'_{(\alpha,\tau)} := \sum_\beta \left| \frac{V'_{\alpha\beta}}{\mathcal{E}_\alpha - \mathcal{E}_\beta + \tau h'_S} \right|^2. \tag{117}$$

The $\chi'_{(\alpha,\tau)}$ have distribution $f'_{FS}$ with asymptotic tail

$$f'_{FS}(\chi') \sim \sqrt{\frac{\chi'_\star}{\chi'^3}}. \tag{118}$$

As $\Delta S$ increases, this tail shifts to larger $\chi$. By direct application of the results of Secs. 4 and 5, $\chi'_{(\alpha,\tau)}$ determines the universal shape of the distribution of entanglement entropies of the second spin at weak and intermediate coupling (80) ($J'^2 \chi'_\star \ll 1$), and the saturation value of two time correlators of the second spin (95). As we have set the second spin to be in the weak coupling regime, there is no corresponding enhancement of the bath felt by the first spin due to the presence of the second spin. If both spins are intermediately coupled, a self consistent treatment is required.

In Fig 12 we numerically verify that the fidelity susceptibilities of the second spin (117) are distributed as (118) with the enhancement to the typical fidelity susceptibility $\chi'_\star = \exp(\Delta S)\, \chi'_\star|_{J=0}$ determined by the matrix element entropy (114). We do this in two equivalent ways one less direct measure with low statistical noise, and one more direct measure with greater statistical noise. In each case we find good agreement with the theoretical prediction. In Fig 12a we plot $\Delta S$ as defined by (111b) with $[|V'_{\alpha\beta}|]$ extracted by diagonalising the spin-ETH model for different values of coupling $J$ to the first spin and averaging over realisations and mid-spectrum states. Statistical error bars are smaller than plot points. The deviation from the theoretical curve is decreasing with $L$. The $\Delta S > \log 2$ seen at small $L$ reflects the deviation from ETH exhibited by particularly small baths.

In Fig 12b we extract $\Delta S$ as defined by (111a) directly from the distribution of fidelity susceptibilities $\chi'_{(\alpha,\tau)}$. We extract the tail coefficient estimate $\chi'_\star(J, \mathcal{E}, h_S)$, in accordance with (118), by aggregating values of $\chi'_{(\alpha,\tau)}$ from the mid-spectrum states of many realisations into a large data set (of size $N$). We sort this sample into descending order $\chi'_1 > \chi'_2 > \ldots > \chi'_N$, and use the identity (derived in App. F)

$$\log \chi'_\star = \frac{1}{M} \sum_{n=1}^{M} \log \chi'_n + 2\log\left(\frac{M}{2eN}\right) + O\left(\frac{M}{N}\right) + O\left(\frac{1}{\sqrt{M}}\right), \tag{119}$$

which holds for any $M \leq N$. The corrections are minimised by restricting the partial sum to the $M = O(N^{2/3})$ largest values, specifically we use $M = \lfloor N^{2/3}/10 \rfloor$. Eq. (111a) converts

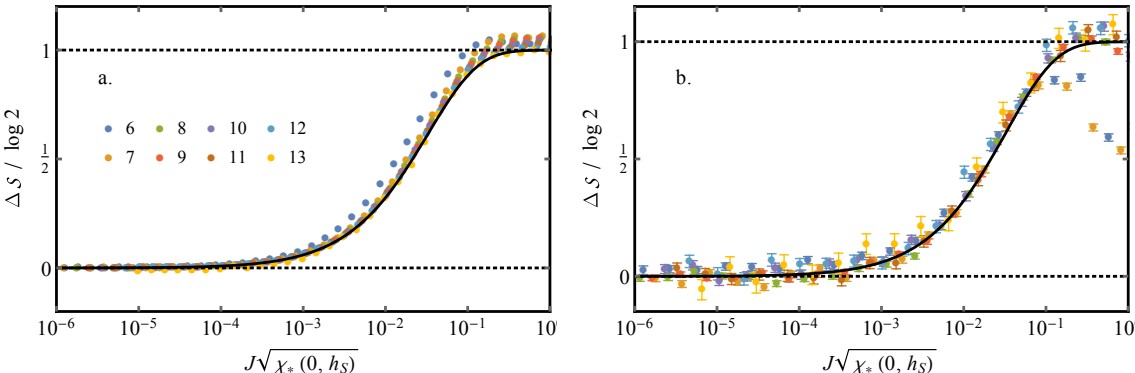

Figure 12: *Entropic enhancement of the bath*: two numerical measures of the entropic enhancement of the bath (coloured points, $L$ values inset) are compared with the theoretical prediction (114) (solid black). Left panel: we extract $\Delta\mathcal{S}$ as defined by (111b). Right panel: we extract $\Delta\mathcal{S}$ as defined by (111a) with $\chi_\star$ extracted using (119). For very small sizes (right panel $L = 6, 7$) there is significant disagreement once the coupling $J$ becomes large. Number of realisations per data point: in the left panel $N = 10, 1000, 2000$ and $N = 6000$ for $L = 13, 12, 11$ and $L \leq 10$ respectively; in the right panel $N = 100, 1000, 3000, 10^4$, and $N = 10^5$ for $L = 13, 12, 11, 10$ and $L \leq 9$ respectively.

the extracted values of $\chi_\star$ into values of the matrix element entropy, $\Delta\mathcal{S}$, which are plotted (coloured points) for different systems size (legend inset). The numerically extracted values of $\Delta\mathcal{S}$ show good agreement with the theoretical prediction (114) (black solid line). The theory curve is calculated using $f_{\mathrm{FS}}(\chi)$ as extracted for the ETH bath in Sec. 3.3, specifically $f_{\mathrm{FS}}$ given by (51b), with $\chi_\star(0, \sigma h_{\mathrm{S}})$ given by (67).

## 8  Discussion

We have developed an ETH-like ansatz of a spin coupled to a finite quantum bath (the Spin-ETH model), this applies in the weak and intermediate regimes where the spin only partially thermalises with the bath. In the intermediate regime, the fraction of states that form many-body resonances determines eigenstate-averaged properties such as the mean spin entanglement entropy, as well as physical observables, such as infinite-time memory and the combined entropy of the spin-bath system as probed by a second spin. Previous analyses of small systems interacting with mesoscopic quantum baths [25, 60, 62–65] overlooked these important effects of many-body resonances.

**Applicability of the two level resonance model:**  Our results hinge on the two level resonance model. It may be surprising that the predictions of this model agree closely with exact-diagonalisation numerics, as it assumes the eigenstates of the Spin-ETH model to be given by a superposition of two eigenstates in the decoupled ($J = 0$) limit,

$$|\mathcal{E}_\alpha\rangle = \sqrt{p_\alpha}|\sigma\rangle|E_a\rangle + \sqrt{q_\alpha}|-\sigma\rangle|E_b\rangle, \tag{120}$$

and estimates the coefficients $p_\alpha, q_\alpha$ within first order degenerate perturbation theory. Accounting for hybridisation with other states at first order, as well as higher order terms, corrects the bath states, and leads to a more refined ansatz

$$|\mathcal{E}_\alpha\rangle = \sqrt{p_\alpha}|\sigma\rangle|\tilde{E}_a\rangle + \sqrt{q_\alpha}|-\sigma\rangle|\tilde{E}_b\rangle. \tag{121}$$

However, providing $J \ll h_{\mathrm{S}}$, the cross term

$$\sqrt{p_\alpha q_\alpha}\langle \tilde{E}_a | \tilde{E}_b \rangle \ll p_\alpha \langle \tilde{E}_a | \tilde{E}_a \rangle, q_\alpha \langle \tilde{E}_b | \tilde{E}_b \rangle \tag{122}$$

is negligible due to conservation of energy. Thus, this more refined ansatz yields the same results as presented in the main text.

**Connections to the many-body localisation finite-size crossover:** Refs. [32,58] found that an ETH-like ansatz (specifically the matrix elements of local operators satisfying (57) but with non-Gaussian $R_{ab}$) applied on the thermal side of the finite-size many-body localisation (MBL) crossover. The authors argued that this violation of Berry's conjecture [54] was a consequence of the sub-diffusive thermalising behaviour.

Our results in Sec. 6 indicate a different origin for the non-Gaussianity unrelated to the presence, or otherwise, of sub-diffusion. Specifically, as each spin is coupled to an external environment made up of the remaining spins, the spin-environment coupling may be characterised by a quantity $J\sqrt{\chi_\star}$. When $J\sqrt{\chi_\star} \lesssim 1$ the spins are in the intermediate coupling regime, and operators on the spin have off diagonal matrix elements which follow a non-Gaussian distribution. The distribution we predict (see Fig. 11, and Sec. 6) reproduces the qualitative features observed in Refs. [32,58].

Non-Gaussianity is present in the intermediate coupling regime of the spin-ETH model, where it appears concurrently with the spin entanglement entropy becoming bi-modally distributed—being either close to the thermal ($S = \log 2$) or localised ($S = 0$) value. Furthermore, the distribution of the $R_{ab}$ will approach a Gaussian when $J\sqrt{\chi_\star}$ is taken sufficiently large $J\sqrt{\chi_\star} \gtrsim 1$. In the context of numerical studies on spin chains, this may be achieved by increasing the system size. Within the disorder regime studied by Refs. [32,58], subsequent analyses have verified both the bi-modal distribution of spin entanglement entropies [66], and tendency towards Gaussian $R_{ab}$ upon increasing system size [67]. This indicates that the thermal side of the numerically observed MBL crossover is in the intermediate coupling regime.

We note that this resonance based explanation of the physics in small system numerics is in line with recent proposals that the numerical MBL-thermal crossover occurs when the MBL phase is destabilised by many-body resonances [68,69], and not by rare thermal regions, as has largely been assumed [23,25,60,61,63,70–72].

**Connections to the Rosenzweig-Porter model:** Our results also connect to the Rosenzweig-Porter (RP) model, though they do not correspond to the well-studied delocalisation transitions [73–77]. Instead, they correspond most closely to RP models in which the typical off-diagonal matrix element and typical level spacing scale together (as $1/d$, where $d$ is the dimension). Thus, within the RP terminology, the intermediate regime of the Spin-ETH model is localised, as the exact eigenstates $|\mathcal{E}_\alpha\rangle$ have significant overlap with only a finite number of the $J = 0$ eigenstates $|\mathcal{E}_\alpha^0\rangle$. However, as we have shown, in the Spin-ETH model this is sufficient to lead to the entropic enhancement of the bath.

**Extensions to this work:** We have focused on the infinite time properties of the system, characterised by eigenstate properties, time averaged correlations, and the properties of the system as an effective bath. It would be interesting to extend our analysis to describe the finite bath corrections to the finite time decay of correlation functions, providing a link between our work and previous random matrix models of decoherence [78–81], and Loschmidt echos [82–88].

A particularly relevant direction for future investigation is extending our analysis to the problem of multiple spins coupled to the bath. We treated the simplified case in Sec. 7 in which the second spin is in the weak coupling regime. However, extension to the case where

the 'effective bath' seen by the second spin is enhanced by the presence of the first spin and *vice versa* is necessary to study the regime where multiple spins are coupled in the intermediate regime.

Moreover, while we have focused on coupling a two-level system, or spin-1/2, to a bath, it would be useful to obtain results for higher dimensional qudits, and even pairs of large weakly coupled baths. The latter case in particular could prove relevant to the RG studies of the MBL transition [89–95], which currently treat pairs of thermal regions as either in the weak or strong coupling regimes. This is a poor approximation at large $d$ where these regimes are asymptotically separated.

Finally, while we have focused on infinite temperature properties of the Spin-ETH system, we expect our results are generalisable to the finite temperature. A subtlety which must be accounted for is the distinct density of states available in the ↑ and ↓ sectors. When this feature is correctly accounted for, at the crossover from the intermediate to strong coupling regimes, the two modes of $f_{EE}(S)$ should combine into a single mode at the thermal entropy $S_{th.} < \log 2$.

## Acknowledgements

We are grateful to F.J. Burnell, S. Gopalakrishnan, C.R. Laumann, S. Morampudi and A. Polkovnikov for useful discussions.

**Funding information**   P.C. is supported by the NSF STC "Center for Integrated Quantum Materials" under Cooperative Agreement No. DMR-1231319, and A.C. is supported by NSF DMR-1813499. Numerics were performed on the Boston University Shared Computing Cluster with the support of Boston University Research Computing Services.

## A   Calculation of $\hat{\rho}_\alpha$ in perturbation theory

In this appendix we provide a step by step derivation of the reduced density matrix in (30) which is calculated to quadratic order in perturbation theory.

Recall the unperturbed Hamiltonian $\mathcal{H}_0$. Consider an arbitrary eigenstate projector of this Hamiltonian

$$\mathcal{P}_\alpha^0 := |\mathcal{E}_\alpha^0\rangle\langle\mathcal{E}_\alpha^0|. \tag{123}$$

Upon introducing a perturbation $\mathcal{H}_0 \to \mathcal{H} = \mathcal{H}_0 + \mathcal{V}$ the perturbed eigenstate projectors are given to infinite order in perturbation theory by

$$\mathcal{P}_\alpha = \sum_{n=0}^{\infty} \mathcal{P}_\alpha^{(n)} = \sum_{n=0}^{\infty} (-1)^{n+1} \sum_{k_j \geq 0\,:\,k_0+k_1+...+k_n=n} \mathcal{S}_\alpha^{(k_0)} \mathcal{V} \mathcal{S}_\alpha^{(k_1)} \mathcal{V} \mathcal{S}_\alpha^{(k_2)} \cdots \mathcal{S}_\alpha^{(k_{n-1})} \mathcal{V} \mathcal{S}_\alpha^{(k_n)}, \tag{124}$$

where the sum is taken over non negative integers $k_j$ such that $\sum_{j=0}^{n} k_j = n$, and we have denoted

$$\mathcal{S}_\alpha^{(0)} = -\mathcal{P}_\alpha^0, \qquad \text{and} \qquad \mathcal{S}_\alpha^{(n>0)} = \mathcal{R}_\alpha^n, \tag{125}$$

where $\mathcal{R}_\alpha$ is the projected resolvent

$$\mathcal{R}_\alpha := \lim_{z \to \mathcal{E}_\alpha^0} (\mathbb{1} - \mathcal{P}_\alpha^0) \frac{1}{\mathcal{H}_0 - z} (\mathbb{1} - \mathcal{P}_\alpha^0) = \sum_{\beta \neq \alpha} \frac{\mathcal{P}_\beta^0}{\mathcal{E}_\beta^0 - \mathcal{E}_\alpha^0}. \tag{126}$$

Eq. (124) is a corollary of the more general results derived in Chapter 2 of Ref. [96], results which are here simplified by restricting to the case that $\mathcal{H}$ is Hermitian and all eigenvalues are non-degenerate (i.e. that each $\mathcal{P}_\alpha^0$ has rank 1).

Writing out the two leading corrections in (124) explicitly we have

$$
\begin{aligned}
\mathcal{P}_\alpha^{(1)} &= -\mathcal{R}_\alpha \mathcal{V} \mathcal{P}_\alpha^0 - \mathcal{P}_\alpha^0 \mathcal{V} \mathcal{R}_\alpha, \\
\mathcal{P}_\alpha^{(2)} &= \mathcal{R}_\alpha \mathcal{V} \mathcal{R}_\alpha \mathcal{V} \mathcal{P}_\alpha + \mathcal{R}_\alpha \mathcal{V} \mathcal{P}_\alpha^0 \mathcal{V} \mathcal{R}_\alpha + \mathcal{P}_\alpha^0 \mathcal{V} \mathcal{R}_\alpha \mathcal{V} \mathcal{R}_\alpha \\
&\quad - \mathcal{R}_\alpha^2 \mathcal{V} \mathcal{P}_\alpha^0 \mathcal{V} \mathcal{P}_\alpha^0 - \mathcal{P}_\alpha^0 \mathcal{V} \mathcal{R}_\alpha^2 \mathcal{V} \mathcal{P}_\alpha^0 - \mathcal{P}_\alpha^0 \mathcal{V} \mathcal{P}_\alpha^0 \mathcal{V} \mathcal{R}_\alpha^2.
\end{aligned}
\tag{127}
$$

As we are interested only in the reduced density matrix on the spin, we will now trace out the bath $\hat{\rho}_\alpha := \mathrm{tr}_E(\mathcal{P}_\alpha)$. In order simplify the explicit expressions we obtain we denote

$$
\chi_\alpha := \sum_b \left| \frac{V_{ab}}{E_a - E_b + \sigma h_S} \right|^2 = O(g^2/J^2), \qquad \chi_\alpha' := \sum_{b \neq a} \left| \frac{V_{ab}'}{E_a - E_b} \right|^2 = O(g^2/J^2).
\tag{128}
$$

We then substitute in form of the interaction $\mathcal{V}$ (23) and simplify. We consider a state $\alpha = (\uparrow, a)$ as in the main text

$$
\mathrm{tr}_E\left(\mathcal{P}_\alpha^0\right) = |\uparrow\rangle\langle\uparrow|,
\tag{129a}
$$

$$
\begin{aligned}
\mathrm{tr}_E\left(\mathcal{P}_\alpha^0 \mathcal{V} \mathcal{R}_\alpha\right) &= J\frac{V_{aa}}{h_S}|\uparrow\rangle\langle\downarrow| \\
&= O\left(\frac{g}{\rho_0 h_S}\right)|\uparrow\rangle\langle\downarrow|,
\end{aligned}
\tag{129b}
$$

$$
\begin{aligned}
\mathrm{tr}_E\left(\mathcal{P}_\alpha^0 \mathcal{V} \mathcal{R}_\alpha \mathcal{V} \mathcal{R}_\alpha\right) &= JJ'\left(\sum_b \frac{V_{ab}V_{ba}'}{h_S(E_b - E_a - h_S)} + \sum_{b \neq a}\frac{V_{ab}'V_{ba}}{h_S(E_b - E_a)}\right)|\uparrow\rangle\langle\downarrow| \\
&= O\left(\frac{g}{\rho_0 h_S}\right)|\uparrow\rangle\langle\downarrow|,
\end{aligned}
\tag{129c}
$$

$$
\begin{aligned}
\mathrm{tr}_E\left(\mathcal{R}_\alpha \mathcal{V} \mathcal{P}_\alpha^0 \mathcal{V} \mathcal{R}_\alpha\right) &= J^2 \sum_b \left|\frac{V_{ab}}{E_a - E_b + h_S}\right|^2 |\downarrow\rangle\langle\downarrow| + J'^2 \sum_{b \neq a}\left|\frac{V_{ab}'}{E_a - E_b}\right|^2 |\uparrow\rangle\langle\uparrow| \\
&\quad + JJ' \sum_{b \neq a}\frac{V_{ab}V_{ba}'}{(E_a - E_b)(E_a - E_b + h_S)}|\downarrow\rangle\langle\uparrow| \\
&\quad + JJ' \sum_{b \neq a}\frac{V_{ab}'V_{ba}}{(E_a - E_b)(E_a - E_b + h_S)}|\uparrow\rangle\langle\downarrow| \\
&= J^2 \chi_\alpha |\downarrow\rangle\langle\downarrow| + J'^2 \chi_\alpha'|\uparrow\rangle\langle\uparrow| + O\left(\frac{g^2}{\rho_0 h_S}\right)|\downarrow\rangle\langle\uparrow| + O\left(\frac{g^2}{\rho_0 h_S}\right)|\uparrow\rangle\langle\downarrow|,
\end{aligned}
\tag{129d}
$$

$$
\begin{aligned}
\mathrm{tr}_E\left(\mathcal{P}_\alpha^0 \mathcal{V} \mathcal{P}_\alpha^0 \mathcal{V} \mathcal{R}_\alpha^2\right) &= \mathrm{tr}_E\left(\mathcal{P}_\alpha^0 \mathcal{V} \mathcal{R}_\alpha^2\right)\mathrm{tr}\left(\mathcal{P}_\alpha^0 \mathcal{V}\right)|\uparrow\rangle\langle\downarrow| \\
&= JJ' \frac{V_{aa}^2}{h_S^2}|\uparrow\rangle\langle\downarrow| \\
&= O\left(\frac{g^2}{\rho_0^2 h_S^2}\right)|\uparrow\rangle\langle\downarrow|,
\end{aligned}
\tag{129e}
$$

$$
\begin{aligned}
\mathrm{tr}_E\left(\mathcal{P}_\alpha^0 \mathcal{V} \mathcal{R}_\alpha^2 \mathcal{V} \mathcal{P}_\alpha^0\right) &= \left(J^2 \sum_b \left|\frac{V_{ab}}{E_a - E_b + h_S}\right|^2 + J'^2 \sum_{b \neq a}\left|\frac{V_{ab}'}{E_a - E_b}\right|^2\right)|\uparrow\rangle\langle\uparrow| \\
&= J^2 \chi_\alpha |\uparrow\rangle\langle\uparrow| + J'^2 \chi_\alpha'|\uparrow\rangle\langle\uparrow|.
\end{aligned}
\tag{129f}
$$

Combining the above terms as in (127) provides an explicit form for $\hat{\rho}_\alpha$ given in (30)

$$\hat{\rho}_\alpha = \begin{pmatrix} 1 - J^2\chi_\alpha & O\left(\frac{g}{\rho_0 h_S}\right) \\ O\left(\frac{g}{\rho_0 h_S}\right) & J^2\chi_\alpha \end{pmatrix} + O\left(\frac{g^2}{\rho_0 h_S}\right) + O(g^3), \tag{130}$$

and hence the entanglement entropy

$$S_\alpha = -\mathrm{tr}(\hat{\rho}_\alpha \log \hat{\rho}_\alpha) = -(1 - J^2\chi_\alpha)\log(1 - J^2\chi_\alpha) - J^2\chi_\alpha \log J^2\chi_\alpha + O\left(\frac{g^2}{\rho_0 h_S}\right) + O(g^3), \tag{131}$$

expanding to leading order yields (31) in the main text.

## B  Calculation of the distribution $f_{\mathrm{FS}}$ for a Poisson bath

In this appendix we provide a step-by-step derivation showing in detail how (39) is obtained from the starting from (36). Our starting point is the definition of the fidelity susceptibility

$$\chi_\alpha = \sum_b \left| \frac{V_{ab}}{E_a - E_b + \sigma h_S} \right|^2, \tag{132}$$

where $\alpha = (a, \sigma)$. In the case of a Poisson bath we may treat each of the energy levels as iid drawn from the density of states, and each of the matrix elements as iid drawn from some distribution. Thus we obtain the cumulant generation function (37)

$$K(t|E, \omega) := d \log\left[ \exp\left( \frac{\mathrm{i}t}{d} \left| \frac{V}{E - E_b + \omega} \right|^2 \right) \right]_{V, E_b}. \tag{133}$$

Writing this out explicitly as an energy integral, and using that $K(t|E, \omega) = K(t, E + \omega, 0)$ to set $\omega = 0$ without loss of generality, we obtain

$$K(t|E, 0) = d \log\left[ \frac{1}{d} \int_{-\infty}^{\infty} \mathrm{d}E' \rho(E') \exp\left( \frac{\mathrm{i}t}{d} \left| \frac{V}{E - E'} \right|^2 \right) \cdot \right]_V. \tag{134}$$

We then change variables to $x = |V|^2/(d|E - E'|^2)$; Taylor expand the density of states about $E$; and collect the $V$ averages. Step by step this gives

$$K(t|E) = d \log\left[ \frac{1}{2d} \int_0^\infty \mathrm{d}x \frac{V e^{\mathrm{i}tx}}{\sqrt{dx^3}} \left\{ \rho\left( E + \frac{V}{\sqrt{xd}} \right) + \rho\left( E - \frac{V}{\sqrt{xd}} \right) \right\} \right]_V \tag{135}$$

$$= d \log\left[ \frac{1}{d} \int_0^\infty \mathrm{d}x \frac{V e^{\mathrm{i}tx}}{\sqrt{dx^3}} \left\{ \rho(E) + O\left( \frac{V^2 \rho''(E)}{xd} \right) \right\} \right]_V \tag{136}$$

$$= d \log\left( \int_0^\infty \mathrm{d}x \, e^{\mathrm{i}tx} \left\{ \frac{[|V|]\rho(E)}{(xd)^{3/2}} + O\left( \frac{[|V|^3]\rho''(E)}{(xd)^{5/2}} \right) \right\} \right). \tag{137}$$

To make further progress we use the following result of Fourier analysis (see e.g. Ref [97])

$$\int_0^\infty \mathrm{d}x \frac{e^{\mathrm{i}tx}}{x^{n+1/2}} = \Gamma(\tfrac{1}{2} - n)(-\mathrm{i}t)^{n-\frac{1}{2}} \quad \text{for} \quad n \in \mathbb{N}, \tag{138}$$

to obtain

$$K(t|E, 0) = d \log\left( 1 - \sqrt{-\frac{4\pi \mathrm{i}\rho(E)^2 [|V|]^2 t}{d^3}} + O\left( \frac{[|V|^3]\rho''(E)t^{3/2}}{d^{5/2}} \right) \right). \tag{139}$$

Above, the unity term in the argument of the logarithm follows from the requirement that $K(t=0|E)=0$. Expanding to leading order provides the desired result (38)

$$K(t|E,0) = -\sqrt{-\frac{4\pi i \rho(E)^2 [|V|]^2 t}{d}} + O\left(\frac{t\rho(E)^2 [|V|]^2}{d}\right).$$ (140)

## C  Calculation of the distribution $f_{\mathrm{FS}}$ for a GUE bath

### C.1  Set-up

In this appendix we adapt the approach of Ref. [39] to calculate the distribution of the fidelity susceptibility $f_{\mathrm{FS}}$, defined in (35), of the fidelity susceptibility, defined in (28).

Specifically we assume the matrix elements $V_{ab}$ in (28) are the elements of a $d \times d$ Gaussian Random matrix $V$ drawn with Dyson index $\beta$. Specifically $V_{ab} \in \mathbb{R}, \mathbb{C}, \mathbb{H}$ for $\beta = 1, 2, 4$ respectively, and the matrix $V$ is drawn from a distribution $\propto \exp\left(-\beta \operatorname{tr}\left(V^2\right)/4\sigma^2\right)$ with $\sigma^2 = 1/d$. The matrix elements of $V$ are Gaussian random numbers with mean $[V_{ab}] = 0$ and two-point correlations

$$[V_{ab} V_{cd}{}^*]_V = \sigma^2 \left(\delta_{ac}\delta_{bd} + \frac{2-\beta}{\beta}\delta_{ad}\delta_{bc}\right).$$ (141)

For now $\beta$ is left general, and we proceed in generality as far as possible, but ultimately we only complete calculation is only in the cases $\beta = 2$. The eigenvalues $E_a$ in (28) are the eigenvalues of a separate random matrix, the "bath hamiltonian" in the main text, here denoted $R$. $R$ is drawn iid from the same distribution as $V$. As the target energy $E_a + \sigma h_{\mathrm{S}}$ is arbitrary, for the purposes of simplifying the calculation we set it to zero. We will discuss afterwards how the result we obtain is generalised to different target energies.

According to the arguments presented in the main text, we expect that at asymptotically large $\chi$ the distribution decays as

$$f_{\mathrm{FS}}(\chi|E,\omega) \sim \frac{\chi_\star^{1/2}}{\chi^{3/2}} \qquad \text{where} \qquad \chi_\star(E,\omega) = \rho(E+\omega)^2 [|V_{ab}|]^2,$$ (142)

with $\rho(E)$ and $a$ given by (15) and (42) respectively.

As the upper tail of $f_{\mathrm{FS}}$, set by $\chi_\star$, flows off to infinity in the limit of large $d$, we will calculate the distribution of the *reduced susceptibility* $x = \chi/\chi_\star$, providing a well behaved large $d$ limit. Specifically we calculate

$$f_{\mathrm{RS}}(x) = \chi_\star f_{\mathrm{FS}}(x\chi_\star|E,-E),$$ (143)

where for simplicity, additionally set $\omega = -E$ so that $E + \omega$ is a mid-spectrum energy, however the calculation below is easily extended to generic energies to obtain the result (142).

### C.2  Calculation of $f_{\mathrm{RS}}(x)$

The distribution of $f_{\mathrm{RS}}(x)$ can be written as

$$\begin{aligned}
f_{\mathrm{RS}}(x) &= \left[\delta\left(x - \frac{\chi}{\chi_\star}\right)\right]_{E,V} = \left[\delta\left(x - \frac{1}{\chi_\star}\sum_b \frac{|V_{ab}|^2}{|E_b|^2}\right)\right]_{E,V} \\
&= \frac{1}{2\pi}\int \mathrm{d}t \left[\exp\left(-\mathrm{i}t\left(x - \frac{1}{\chi_\star}\sum_b \frac{|V_{ab}|^2}{|E_b|^2}\right)\right)\right]_{E,V},
\end{aligned}$$ (144)

where in the final equality we have substituted the integral representation of the $\delta$-function. Performing the integration over the Gaussian distributed matrix elements $V_{ab}$ we obtain

$$
\begin{aligned}
f_{\text{RS}}(x) &= \frac{1}{2\pi} \int dt\, e^{-itx} \left[ \prod_b \left( 1 - \frac{2it\sigma^2}{\chi_\star \beta |E_b|^2} \right)^{-\beta/2} \right]_E \\
&= \frac{1}{2\pi} \int dt\, e^{-itx} \left[ \prod_b \left( \frac{|E_b|^2}{|E_b|^2 - \frac{2it\sigma^2}{\chi_\star \beta}} \right)^{\beta/2} \right]_E,
\end{aligned}
\tag{145}
$$

where $\beta = 1, 2, 4$ for matrix elements $V_{ab}$ which are real, complex and quaternion respectively. We then use that $\det R = \prod_b E_b$ for a Gaussian random matrix $R$, and swap the average over eigenvalues, for an ensemble averaging of $R$

$$
f_{\text{RS}}(x) = \frac{1}{2\pi} \int dt\, e^{-itx} \left[ \left( \frac{\det R^2}{\det \left( R^2 - \frac{2it\sigma^2}{\chi_\star \beta} \right)} \right)^{\beta/2} \right]_R .
\tag{146}
$$

We next use the Gaussian integral result

$$
1 = \left( \frac{\beta}{2\pi} \right)^{\beta/2} \int_{\mathcal{M}} dz_i \exp\left( -\frac{\beta}{2} z_i^* z_i \right),
\tag{147}
$$

where for $\beta = 1, 2, 4$ the integral is over real $\mathcal{M} = \mathbb{R}$, complex $\mathcal{M} = \mathbb{C}$, and quaternion $\mathcal{M} = \mathbb{H}$ respectively. This integral is well known for real and complex $z_i$, and holds also for quaternions [98]. From this relation we obtain

$$
\frac{1}{(\det A)^{\beta/2}} = \left( \frac{\beta}{2\pi} \right)^{d\beta/2} \int_{\mathcal{M}^d} dz \exp\left( -\frac{\beta}{2} z^\dagger A z \right),
\tag{148}
$$

for any positive definite matrix $A$. Inserting (148) into (146) one obtains

$$
\begin{aligned}
f_{\text{RS}}(x) &= \frac{1}{2\pi} \int dt\, e^{-itx} \cdot \left( \frac{\beta}{2\pi} \right)^{d\beta/2} \int_{\mathcal{M}^d} dz\, e^{i|z|^2 t\sigma^2/\chi_\star} \left[ (\det R^2)^{\beta/2} \exp\left( -\frac{\beta}{2} z^\dagger R^2 z \right) \right]_R \\
&= \left( \frac{\beta}{2\pi} \right)^{d\beta/2} \int_{\mathcal{M}^d} dz\, \delta\left( x - |z|^2 \sigma^2/\chi_\star \right) \left[ (\det R^2)^{\beta/2} \exp\left( -\frac{\beta}{2} z^\dagger R^2 z \right) \right]_R,
\end{aligned}
\tag{149}
$$

where in the second line we have performed the $t$ integral. As the ensemble of $R$ is Haar invariant, the integrand depends only on $|z|$, thus we may use the relation perform the angular/phase part of the $z$-integral. Specifically:

$$
\begin{aligned}
\int_{\mathcal{M}^d} dz \cdot g(|z|) &= \int_0^\infty r^{d\beta-1} dr \cdot \int d\Omega \cdot g(r) = S_{d\beta-1} \cdot \int_0^\infty r^{d\beta-1} dr \\
&= \frac{2 \cdot \pi^{d\beta/2}}{\Gamma(d\beta/2)} \int_0^\infty r^{d\beta-1} dr \cdot g(r),
\end{aligned}
\tag{150}
$$

where $S_n = 2\pi^{(n+1)/2}/\Gamma(\frac{n+1}{2})$ is the surface are of an $n$-sphere, which lives in $n+1$ dimensional space. Using (150) to simplify (149) we obtain

$$
\begin{aligned}
f_{\text{RS}}(x) &= \frac{2(\beta/2)^{d\beta/2}}{\Gamma(d\beta/2)} \int_0^\infty dr\, r^{d\beta-1} \delta\left( x - r^2\sigma^2/\chi_\star \right) \left[ (\det R^2)^{\beta/2} \exp\left( -\frac{\beta}{2} r^2 u^\dagger R^2 u \right) \right]_R \\
&= \frac{2(\beta/2)^{d\beta/2}}{\Gamma(d\beta/2)} \cdot \frac{\left( x\chi_\star/\sigma^2 \right)^{d\beta/2}}{2x} \cdot \left[ (\det R^2)^{\beta/2} \exp\left( -\frac{\beta x \chi_\star}{2\sigma^2} u^\dagger R^2 u \right) \right]_R,
\end{aligned}
\tag{151}
$$

where is $u$ is an arbitrary fixed unit vector which we set to $u = (1, 0, 0, \cdots)$, and in the second line we have then subsequently performed the radial integral.

To make further progress we decompose $R$ into: a scalar $y \in \mathbb{R}$, a $d-1$ element vector $v \in \mathcal{M}^{d-1}$ and a $(d-1) \times (d-1)$ random matrix $R'$, which is of the same symmetry class as $R$

$$R = \begin{bmatrix} y & v^\dagger \\ v & R' \end{bmatrix}. \tag{152}$$

We may correspondingly decompose the average over $R$ into and average over $y, v, R'$

$$
\begin{aligned}
[g(R)]_R &= \frac{1}{Z} \int dR\, e^{-\frac{\beta}{4\sigma^2}\mathrm{tr}(R^2)} \cdot g(R) \\
&= \frac{1}{Z} \int dR'\, e^{-\frac{\beta}{4\sigma^2}\mathrm{tr}(R'^2)} \cdot \int dv\, e^{-\frac{\beta}{2\sigma^2}v^\dagger v} \cdot \int dy\, e^{-\frac{\beta}{4\sigma^2}y^2} \cdot g\left(\begin{bmatrix} y & v^\dagger \\ v & R' \end{bmatrix}\right) \\
&= \left[ g\left(\begin{bmatrix} y & v^\dagger \\ v & R' \end{bmatrix}\right) \right]_{y,v,R'},
\end{aligned} \tag{153}
$$

where $Z$ is a normalisation constant. In addition we use the relation

$$\det R^2 = \det R'^2 \left( y - v^\dagger R'^{-1} v \right)^2. \tag{154}$$

Inserting (152), (153), (154) into (151) we then obtain

$$
\begin{aligned}
f_{\mathrm{RS}}(x) = \frac{2(\beta/2)^{d\beta/2}}{\Gamma(d\beta/2)} \cdot \frac{\left(x\chi_\star/\sigma^2\right)^{d\beta/2}}{2x} \\
\times \left[ (\det R'^2)^{\beta/2} \left| y - v^\dagger R'^{-1} v \right|^\beta \exp\left(-\frac{\beta x \chi_\star}{2\sigma^2}(y^2 + v^\dagger v)\right) \right]_{y,vR'}.
\end{aligned} \tag{155}
$$

The exponential terms in (155) can be scaled out by using the property

$$\left[ f(y) e^{-a y^2} \right]_y = \left[ \frac{1}{\sqrt{1 + 4a\sigma^2/\beta}} f\left(\frac{y}{\sqrt{1 + 4a\sigma^2/\beta}}\right) \right]_y, \tag{156}$$

which is obtained using the substitution $y \to y' = y\sqrt{1 + 4a\sigma^2/\beta}$, and similarly

$$\left[ f(v) e^{-a v^\dagger v} \right]_v = \left[ \frac{1}{(1 + 2a\sigma^2/\beta)^{\beta(d-1)/2}} f\left(\frac{v}{\sqrt{1 + 2a\sigma^2/\beta}}\right) \right]_v. \tag{157}$$

Using (156) and (157) to simplify (155) we obtain

$$
\begin{aligned}
f_{\mathrm{RS}}(x) &= \frac{2(\beta/2)^{d\beta/2}}{\Gamma(d\beta/2)} \cdot \frac{\left(x\chi_\star/\sigma^2\right)^{d\beta/2}}{2x} \cdot \frac{1}{\sqrt{1 + 2x\chi_\star}} \cdot \frac{1}{(1 + x\chi_\star)^{\beta(d-1)/2}} \\
&\quad \times \left[ (\det R'^2)^{\beta/2} \cdot \left| \frac{y}{\sqrt{1 + 2x\chi_\star}} - \frac{v^\dagger R'^{-1} v}{1 + x\chi_\star} \right|^\beta \right]_{y,vR'} \\
&= \frac{(d\beta/2)^{d\beta/2}}{\Gamma(d\beta/2)} \cdot \frac{1}{x\sqrt{1 + 2x c_\beta d/\pi^2}} \cdot \frac{\left(x c_\beta d/\pi^2\right)^{d\beta/2}}{(1 + x c_\beta d/\pi^2)^{\beta(d-1)/2}} \\
&\quad \times \left[ (\det R'^2)^{\beta/2} \cdot \left| \frac{y}{\sqrt{1 + 2x c_\beta d/\pi^2}} - \frac{v^\dagger R'^{-1} v}{1 + x c_\beta d/\pi^2} \right|^\beta \right]_{y,vR'},
\end{aligned} \tag{158}
$$

where in the second line we have simply subsititituted $\sigma^2 = 1/d$ and $\chi_\star = c_\beta d/\pi^2$ We can simplify this slightly by noting that in the limit of large $d$

$$\frac{(xc_\beta d/\pi^2)^{\beta(d-1)/2}}{(1+xc_\beta d/\pi^2)^{\beta(d-1)/2}} \sim \exp\left(-\frac{\beta\pi^2}{2c_\beta x}\right) \tag{159}$$

and by Stirling's formula

$$\Gamma(d\beta/2) \sim \sqrt{\frac{4\pi}{d\beta}}\left(\frac{d\beta}{2e}\right)^{d\beta/2}, \tag{160}$$

where in all cases $\sim$ denotes asymptotic equality in the limit of large $d$. Thus

$$f_{\mathrm{RS}}(x) \sim \frac{e^{d\beta/2}}{\sqrt{4\pi}} \cdot \frac{1}{x\sqrt{2xc_\beta d/\pi^2}} \cdot \exp\left(-\frac{\beta\pi^2}{2c_\beta x}\right)\left[(\det R'^2)^{\beta/2} \cdot \left|\frac{y}{\sqrt{2}} - \frac{v^\dagger R'^{-1}v}{\sqrt{xc_\beta d/\pi^2}}\right|^\beta\right]_{y,vR'}. \tag{161}$$

As argued in the main text, large values of $\chi_\alpha$ are dominated by the "most resonant" term in the sum. To make this statement precise, let

$$R_\alpha := \left|\frac{V_{ab}}{E_a - E_b + \sigma h_{\mathrm{S}}}\right|^2, \tag{162}$$

where $\alpha = (\sigma, a)$ and $b$ is chosen as to minimise the denominator. Exactly analogous to (35) we define the distribution of this quantity as

$$f_R(R|E, \sigma h_{\mathrm{S}}) := \frac{[\delta(R - R_\alpha)\delta(E - E_a)]_{\mathrm{B}}}{[\delta(E - E_a)]_{\mathrm{B}}}, \tag{163}$$

which is related precisely to $f_{\mathrm{FS}}$ by

$$\lim_{\chi\to\infty} \frac{f_{\mathrm{FS}}(\chi|E, \sigma h_{\mathrm{S}})}{f_R(\chi|E, \sigma h_{\mathrm{S}})} = 1. \tag{164}$$

From this it follows, by the arguments in the main text, that

$$f_R(\chi) \sim \sqrt{\frac{\chi_\star}{\chi^3}}, \tag{165}$$

and thus

$$f(x) \sim x^{-3/2}. \tag{166}$$

Using (166) to simplify the $x$-independent constants in (161) we find

$$f_{\mathrm{RS}}(x) \sim \frac{1}{x^{3/2}} \cdot \exp\left(-\frac{\beta\pi^2}{2c_\beta x}\right) \frac{\left[(\det R'^2)^{\beta/2} \cdot \left|y - \frac{v^\dagger R'^{-1}v}{\sqrt{xc_\beta d/(\pi^2\sqrt{2})}}\right|^\beta\right]_{y,vR'}}{\left[(\det R'^2)^{\beta/2} \cdot |y|^\beta\right]_{y,vR'}}. \tag{167}$$

To make further progress we consider the cases $\beta = 1, 2, 4$ individually.

### C.2.1 $f_{\mathrm{FS}}$ for GUE

The simplest case is GUE matrices ($\beta = 2$). Expanding the quadratic in (167), noting that the cross term, which is odd in $y$ thus integrates to zero, and substituting $c_{\beta=2} = \pi/4$ one finds

$$f_{\mathrm{RS}}(x) = \exp\left(-\frac{4\pi}{x}\right) \cdot \frac{1}{x^{3/2}} \cdot \left(1 + \frac{8\pi}{x}\right), \tag{168}$$

where the coefficient $8\pi$ on the second term in the brackets is determined by enforcing that the distribution is normalised $\int \mathrm{d}x\, f_{\mathrm{RS}}(x) = 1$.

### C.2.2  $f_{FS}$ for GSE

Following the same approach for ($\beta = 4$), expanding (167) and performing the $y$-integrals, and substituting $c_{\beta=4} = 9\pi/32$ one finds

$$f_{RS}(x) = \exp\left(-\frac{9\pi}{64x}\right) \cdot \frac{1}{x^{3/2}} \cdot \left(1 + \frac{C}{x} + \frac{C'}{x^2}\right), \tag{169}$$

where by normalisation we determine that $8192 C' + 768 C\pi + 135\pi^2 = 0$. However this leaves the remaining degree of freedom undetermined. Unfortunately we have been unable to determine the values of $C, C'$ exactly.

### C.2.3  $f_{FS}$ for GOE

For GOE ($\beta = 1$), we set $c_{\beta=1} = 2/\pi$, however the terms inside the brackets are not easily expanded

$$f_{RS}(x) = \exp\left(-\frac{\pi^3}{4x}\right) \cdot \frac{1}{x^{3/2}} \cdot \frac{\left[\left|\det R'\right| \cdot \left|y - \frac{v^\dagger R'^{-1} v}{\sqrt{x d \sqrt{2}/\pi^3}}\right|\right]_{y, vR'}}{\left[\left|\det R'\right| \cdot |y|\right]_{y, vR'}}, \tag{170}$$

however by performing the $y-$integral we obtain

$$f_{RS}(x) = \exp\left(-\frac{\pi^3}{4x}\right) \cdot \frac{1}{x^{3/2}} \left(1 + \frac{\left[\left|\det R'\right| \cdot g\left(\frac{v^\dagger R'^{-1} v}{\sqrt{x \sqrt{2}/\pi^3}}\right)\right]_{v, R'}}{\left[\left|\det R'\right|\right]_{v, R'}}\right), \tag{171}$$

where $g(z) = e^{-z^2/4} - 1 + (\sqrt{\pi} z/2) \operatorname{Erf}(z/2)$. As we expect the $R'$ average to be dominated by the cases where $R'$ is close to singular, (i.e. $|R^{-1}|$ large), in which regime $g(z) \propto |z| + O(z^0)$, we anticipate that the sub-leading terms come in powers of $x^{-1/2}$:

$$f_{RS}(x) = \exp\left(-\frac{\pi^3}{4x}\right) \cdot \frac{1}{x^{3/2}} \left(1 + \frac{C}{x^{1/2}} + \frac{C'}{x} + \dots\right). \tag{172}$$

## D  Fermi's Golden Rule

In this appendix we show that Fermi's Golden rule (FGR) predicts an exponential decay of the infinite temperature correlator two-time connected correlator

$$C_{zz}(t) := \operatorname{tr}\left(e^{i\mathcal{H}t} \sigma^z e^{-i\mathcal{H}t} \sigma^z \hat{\varrho}_0\right) = e^{-\gamma t}. \tag{173}$$

The calculation is a little more complex than simply resolving the trace over the initial states $|\mathcal{E}_\alpha^0\rangle$ and asserting that each one has an amplitude which is decaying at the FGR rate. By conservation of probability one must also consider the influx of amplitude generated by states from the opposite spin sector, this correction leads to an $O(1)$ pref factor on the FGR.

The decay rate we calculate in this section sets the exponential decay of correlations. We note that the same value of $\gamma$ is obtained for a treatment of the spin dynamics using the Lindblad equation of motion.

To apply FGR we first rearrange the correlator into the form

$$C_{zz}(t) = \left(\sum_{\sigma\tau} \sigma\tau P_{\sigma|\tau} P_\tau\right) - \left(\sum_{\sigma\tau} \sigma P_{\sigma|\tau} P_\tau\right)\left(\sum_\tau \tau P_\tau\right), \tag{174}$$

where the sum is over $\sigma, \tau \in \{\uparrow, \downarrow\}$ where $\uparrow, \downarrow$ are taken to have numerical values $+1, -1$ respectively, and the probabilities are given by the expectation values

$$P_\sigma = \langle \Pi_\sigma(0) \rangle_{\hat{\varrho}_0}, \tag{175a}$$

$$P_{\sigma|\tau}(t) = \langle \Pi_\sigma(t) \Pi_\tau(0) \rangle_{\hat{\varrho}_0} / \langle \Pi_\tau(0) \rangle_{\hat{\varrho}_0}, \tag{175b}$$

where $\Pi_\sigma(t)$ is the projector onto a spin sector in the Heisenberg picture. By rearranging (174) is recast as

$$C_{zz}(t) = 1 - P_{\uparrow|\downarrow}(t) - P_{\downarrow|\uparrow}(t). \tag{176}$$

To apply Fermi's Golden rule we decompose this into their different energy contributions $P_{\sigma|\tau}(t) = \int dE \, p_{\sigma|\tau}(t, E)$ where $p_{\sigma|\tau}(t, E) dE$ is the probability that the spin is in state $\sigma$ with bath energy in the range $[E, E + dE]$, given the boundary condition $p_{\sigma|\tau}(0, E) = \delta_{\sigma\tau} \rho(E)/d$. Fermi's golden rule states that

$$\partial_t p_{\sigma|\tau}(t, E) = \Gamma_{-\sigma}(E + \sigma h_S) p_{-\sigma|\tau}(t, E + \sigma h_S) - \Gamma_\sigma(E) p_{\sigma|\tau}(t, E), \tag{177}$$

where the decay rate $\Gamma_\sigma(E) = 2\pi J^2 \tilde{v}(E, \sigma h_S)$ is determined by (87), and the two terms respectively account for the decays of $-\sigma$ states into the $\sigma$ sector and *vice versa*. The solution is given by

$$p_{\sigma|-\sigma}(t, E) = \frac{\rho(E + \sigma h_S)}{2d} \left(1 - \frac{\Gamma_\sigma^-(E)}{\Gamma_\sigma^+(E)}\right) \left(1 - e^{-\Gamma_\sigma^+(E)t}\right), \tag{178}$$

where we have denoted $\Gamma_\sigma^\pm(E) = \Gamma_\sigma(E) \pm \Gamma_{-\sigma}(E + \sigma h_S)$ (note $\Gamma_\sigma^\pm(E - \sigma h_S) = \pm \Gamma_{-\sigma}^\pm(E)$). Thus we obtain

$$
\begin{aligned}
C_{zz}(t) &= 1 - \frac{1}{2d} \sum_\sigma \int dE \rho(E + \sigma h_S) \left(1 - \frac{\Gamma_\sigma^-(E)}{\Gamma_\sigma^+(E)}\right) \left(1 - e^{-\Gamma_\sigma^+(E)t}\right) \\
&= 1 - \frac{1}{2d} \sum_\sigma \int dE \rho(E) \left(1 + \frac{\Gamma_\sigma^-(E)}{\Gamma_\sigma^+(E)}\right) \left(1 - e^{-\Gamma_\sigma^+(E)t}\right).
\end{aligned}
\tag{179}
$$

Expanding $\log C_{zz}(t)$ in powers of $t$ we obtain

$$
\begin{aligned}
\log C_{zz}(t) &= \sum_n \frac{\kappa_n t^n}{n!} \\
&= C_{zz}'(0)t + \frac{1}{2}\left(C_{zz}''(0) - C_{zz}'(0)^2\right)t^2 \\
&\quad + \frac{1}{6}\left(C_{zz}'''(0) - 3C_{zz}''(0)C_{zz}'(0) + 2C_{zz}'(0)^3\right)t^3 + \dots,
\end{aligned}
\tag{180}
$$

where

$$
\begin{aligned}
\kappa_1 &= C_{zz}'(0), \\
\kappa_2 &= \left(C_{zz}''(0) - C_{zz}'(0)^2\right), \\
\kappa_3 &= \left(C_{zz}'''(0) - 3C_{zz}''(0)C_{zz}'(0) + 2C_{zz}'(0)^3\right), \\
&\;\;\vdots
\end{aligned}
\tag{181}
$$

One finds $\kappa_1 = O(L^0)$, whereas higher order terms are suppressed, this follows as the density of states $\rho(E)$ is asymptotically narrower than the scale on which $\Gamma_\sigma(E)$ varies, specifically, $\kappa_2 = O(L^{-1})$ and $\kappa_{n>2} = O(L^{-n})$. We may thus neglect the sub-leading terms in the large system limit. Thus we have

$$\log C_{zz}(t) = -\gamma t + O(t^2/L), \tag{182}$$

where

$$\gamma = -C'_{zz}(0) = \frac{1}{2d}\sum_\sigma \int dE\rho(E)\big(\Gamma^+_\sigma(E) + \Gamma^-_\sigma(E)\big) = \frac{2\pi J^2}{d}\sum_\sigma \int dE\rho(E)\tilde{v}(E,\sigma h_S), \quad (183)$$

which is the value (90) quoted in the main text. For the Spin-ETH model studied in the main text we find numerically

$$\gamma = J^2 \times 1.64\ldots. \qquad (184)$$

## E Asymptotic form of the matrix element entropy

In this appendix we show the matrix element entropy has the limiting small $J$ behaviour

$$\Delta S(J,\mathcal{E},h'_S) \sim -8J\sqrt{\chi_\star(\mathcal{E},h'_S)}\log\left(J\sqrt{\chi_\star(\mathcal{E},h'_S)}\right) \qquad (185)$$

given as (116a) in the main text. Here and throughout this section $\sim$ is used to denote asymptotic equality, and we assume we have already taken the limit of large dimension $d \to \infty$ while holding $\chi_\star$ fixed i.e. $J^2\chi_\star$ may be tuned arbitrarily small without leaving the intermediate regime. Here the matrix element entropy is defined by

$$\Delta S := 2\log\left(\int d\chi \int d\chi' f_{FS}(\chi)f_{FS}(\chi')K(J^2\chi, J^2\chi')\right), \qquad (186a)$$

$$K(x,y) := \sqrt{p(x)p(y) + q(x)q(y)} + \sqrt{p(x)q(y) + q(x)p(y)}, \qquad (186b)$$

$$p(x) := 1 - q(x) := \frac{1}{2}\left(1 - \frac{1}{\sqrt{1+4x}}\right). \qquad (186c)$$

(114) in the main text, where for brevity we have suppressed dependency on $\mathcal{E}, h'_S$.

In the limit of $J \to 0$ the integral converges to unity, and hence $\Delta S = 0$. It is useful to separate off this limiting value

$$\Delta S = 2\log\left(1 + \frac{1}{2}I\right) = I + O(I)^2, \qquad (187a)$$

$$I := 2\int d\chi \int d\chi' f_{FS}(\chi)f_{FS}(\chi')\big(K(J^2\chi, J^2\chi') - 1\big). \qquad (187b)$$

We then proceed by making a substitution $s = 2\sqrt{\chi_\star/\chi}$ to obtain

$$I = 2\int_0^\infty ds \int_0^\infty ds' f_s(s)f_s(s')\left(K\left(\frac{4J^2\chi_\star}{s^2}, \frac{4J^2\chi_\star}{s'^2}\right) - 1\right), \qquad (188)$$

where the distribution of the $s$ is given by

$$f_s(s) = f_\chi\left(\frac{4\chi_\star}{s^2}\right)\cdot\left|\frac{d\chi}{ds}\right| = 1 + O(s) \qquad (189)$$

and decaying $-\log f(s) \sim s^2$ at large $s$.

Consider the integral $I$, we note two properties of its integrand $K-1$: (i) in the limit of small $J$ the integrand $K-1$ tends to zero everywhere except for the neighbourhood of the lines $s = 0$ and $s' = 0$; (ii) in the limit of small $J$ the derivative $\partial_s K$ is non zero only in the neighbourhood of $s = 0$, and similarly the derivative $\partial_{s'} K$ is non zero only in the neighbourhood of $s' = 0$. With these properties, one can see that the small $J$ limit of $I$ is the same for any choice of

distribution $f_s(s)$ which is smooth in the vicinity of 0, and preserves the value of $f_s(0)$. As a result we are at liberty to choose a much "nicer" distribution to work with. We choose

$$f_s(s) = \begin{cases} 1 & \text{for} \quad s \in [0,1] \\ 0 & \text{otherwise} \end{cases} \tag{190}$$

to obtain

$$I \sim I' := 2\int_0^1 \mathrm{d}s \int_0^1 \mathrm{d}s' \left( K\left( \frac{4J^2\chi_\star}{s^2}, \frac{4J^2\chi_\star}{s'^2} \right) - 1 \right). \tag{191}$$

From here we continue by substituting $p = p(4J^2\chi_\star/s^2)$ and $p_0 = p(4J^2\chi_\star)$ to obtain

$$I' = 2\int_{p_0}^{1/2} \mathrm{d}p \int_{p_0}^{1/2} \mathrm{d}p' f_p(p) f_p(p') \left( \sqrt{pp'+(1-p)(1-p')} + \sqrt{p(1-p')+p'(1-p)} - 1 \right). \tag{192}$$

Where the distribution of $p$ is given by

$$f_p(p) = \left| \frac{\mathrm{d}s}{\mathrm{d}p} \right| = \frac{J\sqrt{\chi_\star}}{p^{3/2}(1-p)^{3/2}} \tag{193}$$

and we have set

$$p_0 = 4J^2\chi_\star + O(J^4\chi_\star^2). \tag{194}$$

We then consider the limit $c := \lim_{p_0 \to 0} I'(p_0^{1/2} \log p_0^{1/2})$ writing $q := 1-p$, $q' := 1-p'$, $q_0 := 1-p_0$ for brevity

$$c = \lim_{p_0 \to 0} \frac{1/2}{p_0^{-1/2}\log(p_0^{1/2})} \int_{p_0}^{1/2} \mathrm{d}p \int_{p_0}^{1/2} \mathrm{d}p' \left( \frac{\sqrt{pp'+qq'}+\sqrt{pq'+qp'}-1}{p^{3/2}q^{3/2}p'^{3/2}q'^{3/2}} \right) \tag{195a}$$

$$= \lim_{p_0 \to 0} \frac{2}{p_0^{-3/2}\log(p_0^{1/2})} \int_{p_0}^{1/2} \mathrm{d}p \left( \frac{\sqrt{pp_0+qq_0}+\sqrt{pq_0+p_0q}-1}{p^{3/2}q^{3/2}p_0^{3/2}q_0^{3/2}} \right) \tag{195b}$$

$$= \lim_{p_0 \to 0} \frac{2}{p_0^{-5/2}\log(p_0^{1/2})} \int_1^{1/(2p_0)} \mathrm{d}r \left( \frac{\sqrt{rp_0^2+(1-rp_0)q_0}+\sqrt{rp_0q_0+p_0(1-rp_0)}-1}{r^{3/2}p_0^{3/2}(1-rp_0)^{3/2}p_0^{3/2}q_0^{3/2}} \right) \tag{195c}$$

$$= \lim_{p_0 \to 0} \frac{2}{p_0^{-5/2}\log(p_0^{1/2})} \int_1^{1/(2p_0)} \mathrm{d}r \left( \frac{\sqrt{1+r}}{r^{3/2}p_0^{5/2}} \right) \tag{195d}$$

$$= \lim_{p_0 \to 0} \frac{2}{p_0^{-5/2}\log(p_0^{1/2})} \cdot \frac{\log p_0}{p_0^{5/2}} \tag{195e}$$

$$= 4, \tag{195f}$$

where: in the second line we have applied l'Hôpitals rule, differentiating with respect to $p_0$; in the third line substituted $p = rp_0$; in the fourth line expanded the integrand to leading order term in $p_0$; in the fifth line performed the integral and kept the result to leading order in $p_0$ before taking the limit.

Combining (187), (191), (194) and (195) we obtain the desired result in the limit of small $J$

$$\Delta S \sim I \sim I' \sim 4\sqrt{p_0}\log\sqrt{p_0} \sim 8J\sqrt{\chi_\star}\log J\sqrt{\chi_\star}. \tag{196}$$

# F    Estimator for $\chi_\star$

In this appendix we give a statistical estimator for obtaining $\chi_\star$ from a sample of $N$ values $\chi_\alpha$ drawn iid from the distribution $f_{\mathrm{FS}}$. Specifically we show that

$$\log \chi_\star = \frac{1}{M} \sum_{n=1}^{M} \log \chi_n + 2 \log \left( \frac{M}{2eN} \right) + O\left( \frac{M}{N} \right) + \left( \frac{1}{\sqrt{M}} \right), \tag{197}$$

where $\chi_1 > \chi_2 > \ldots > \chi_N$ are the rank ordered $\chi_\alpha$, and setting $M = O(N^{2/3})$ minimises the sub-leading corrections. This estimator has asymptotic error $O(N^{2/3})$ which we believe may be the minimum possible asymptotic error.

The $\chi_\alpha$ are drawn from the distribution $f_{\mathrm{FS}}$, which is given to leading and first sub-leading order by

$$f_{\mathrm{FS}}(\chi) = \frac{\chi_\star^{1/2}}{\chi^{3/2}} + a \frac{\chi_\star}{\chi^2} + O\left( \frac{\chi_\star^{3/2}}{\chi^{5/2}} \right). \tag{198}$$

Consider the quantities $v_1 < v_2 < \ldots < v_N$ defined by

$$v_n = \left( \frac{\chi_n^{1/2}}{2\chi_\star^{1/2}} - \frac{c}{4} \right)^{-1}. \tag{199}$$

The $v_n$ are distributed according to

$$f_v(v) = f_{\mathrm{FS}}(\chi) \cdot \left| \frac{\mathrm{d}\chi}{\mathrm{d}v} \right| = 1 + O(v^2). \tag{200}$$

Intuitively, in the vicinity of $v = 0$ the distribution $f_v$ behaves like the uniform distribution

$$f_u(u) = \begin{cases} 1 & \text{for} \quad u \in [0, 1] \\ 0 & \text{otherwise} \end{cases}. \tag{201}$$

This can be made precise in the sense of the following result

$$\left[ \frac{1}{M} \sum_{n=1}^{M} \log v_n \right] = \left[ \frac{1}{M} \sum_{n=1}^{M} \log u_n \right] + O\left( \frac{M^2}{N^2} \right), \tag{202}$$

where the $u_1 < u_2 < \ldots < u_N$ are a rank ordered sample of values drawn iid from $f_u$.

Using (202) to relate to expectation values of calculated under the uniform distribution is useful, as it is significantly more simple to work with. In particular the mariginal distribution of the smallest $M$ values of a sample of size $N$ is given by

$$f_{u,M}(u) = \sum_{n=1}^{M} \frac{N!}{M(n-1)!(N-n)!} u^{n-1} (1-u)^{N-n}, \tag{203}$$

(this is a standard result of Order statistics, see for example, Section 5.4 of Ref. [99]) from which it is readily calculated that

$$\left[ \frac{1}{M} \sum_{n=1}^{M} \log u_n \right] = \int_0^1 \log u \, f_{u,M}(u) \mathrm{d}u = H_M - H_N - 1 = \log \left( \frac{M}{N} \right) + O\left( \frac{1}{N} \right), \tag{204}$$

where $H_n = \sum_{k=1}^{n} 1/k = \gamma + \log n + O(1/n)$ is the $n$th harmonic number, and $\gamma$ the Euler-Mascheroni constant. Lastly we note that while (204) describes the ensemble averaged value, for any individual sample there will additionally be statistical noise

$$\frac{1}{M}\sum_{n=1}^{M}\log u_n = \left[\frac{1}{M}\sum_{n=1}^{M}\log u_n\right] + O\left(\frac{1}{\sqrt{M}}\right). \tag{205}$$

We are now able to arrive at our desired result

$$\frac{1}{M}\sum_{n=1}^{M}\log \chi_n = \left[\frac{1}{M}\sum_{n=1}^{M}\log \chi_n\right] + O\left(\frac{1}{\sqrt{M}}\right) \tag{206a}$$

$$= \log \chi_\star + \left[\frac{2}{M}\sum_{n=1}^{M}\log\left(\frac{2}{v_n}+\frac{c}{2}\right)\right] + O\left(\frac{1}{\sqrt{M}}\right) \tag{206b}$$

$$= \log \chi_\star + 2\log 2 - \left[\frac{2}{M}\sum_{n=1}^{M}\log v_n\right] + O\left(\frac{1}{M}\left[\sum_{n=1}^{M}v_n\right]\right) + O\left(\frac{1}{\sqrt{M}}\right) \tag{206c}$$

$$= \log \chi_\star + 2\log 2 - \left[\frac{2}{M}\sum_{n=1}^{M}\log u_n\right] + O\left(\frac{M}{N}\right) + O\left(\frac{1}{\sqrt{M}}\right) \tag{206d}$$

$$= \log \chi_\star + 2\log 2 - 2\log\left(\frac{M}{N}\right) + 2 + O\left(\frac{M}{N}\right) + O\left(\frac{1}{\sqrt{M}}\right). \tag{206e}$$

Where in the second line we have substituted $v_n$ (199); in the third line we have expanded the argument of the logarithm in powers of $c$; in the fourth line we have substituted Eq. (202) and evaluated the summation in the correction term; in the fifth line we have substituted Eq. (204). It is then a matter of simple rearrangement to obtain Eq. (197).

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
