# Peer review of "Partial thermalisation of a two-state system coupled to a finite quantum bath"

_SciPost Physics, doi:SciPost Phys. 12, 103 (2022)_

## Round 1 · Referee Report · Anonymous · 2021-7-8

Strengths

1- Detailed analysis of the distribution of the susceptibility
2- Basing the analysis of other observables (entropy, asymptotic values of spin evolution) on the distribution of the susceptibility
3- Finite size scaling discussion allows comparison with numerics and experiments

Weaknesses

1- Too short discussion of the implications for the MBL phase/transition
2- (minor point) Very technical on some derivations, whose analysis feels like it could have been simplified

Report

Referee report on "Partial thermalisation of a two-state system coupled to a
finite quantum bath" by P. J. D. Crowley, A. Chandran.

The paper studies the different coupling regimes of a single spin to a bath, the latter described by either one of the Gaussian ensembles or a random matrix with Poisson statistics or a realistic model given by an ergodic spin chain.

The main object of study is the susceptibility $\chi$ in Eq. (3) or (28).

This is a random variable good to study the response of a system to the introduction of a local operator, which has been studied by several people before. In the specific case at hand, I think the authors fail to recognize that this is exactly the first term in a locator expansion in the spirit of [1] and [2] and more recently in several other papers (see the more recent [3]).
In [1-3] the denominators in the energy are uncorrelated, while in principle $E_a, E_b$ in this paper are, for some ensembles, correlated. But the long tail
$$
f\sim \chi^{-3/2},
$$
which is a main feature of the distribution, follows from small denominators. The small denominators come from pairs $a,b$ for which $E_a-E_b\sim h_s=O(1),$ but since the level spacing $\delta \ll 1\sim h_S$ (which is the scale on which $E_a,E_b $ correlate), this means that $E_a$ and $E_b$ are uncorrelated to a very good approximation. So we fall back in the conditions of [1,2], and the tail follows from the discussion after eq. (5.1) in [2]. Analogous results are in the denominator analysis in [4,5].

I think the authors should recognize this fact, which does not subtract to their analysis (which goes beyond this point, significantly) but it connects with a set of papers which the authors have not involved in their discussion. I was in particular surprised of the absence of reference to [4] in their otherwise very generous bibliography, considering this work has significant implications for MBL, for which the authors have a separate sub-section.

Regarding the issue of whether ETH should be modified or not in the thermalizing region preceding the MBL transition is not clear to me if the authors agree or not with their references (in the paper [32,53]) and if they agree or disagree with [6] (in this report) which seems to find that if one goes to sufficiently large system sizes, the distribution of the off-diagonal matrix elements return gaussian and that this is not in contradiction with having subdiffusive transport.

For the rest of the paper I have no objections or comments. I like the treatment of the off-diagonal elements of the operators for the intermediate values of couplings, I think it is an original and nice addition to the ETH vulgata and it deserves to be published. The paper as a whole deserves publication, once the authors fix the discussion above.

I caught only one typo:

Fig. 12. Upper and lower panel should be left and right panel in the caption.

[1] Anderson, P. W. (1958). Absence of diffusion in certain random lattices. Physical review, 109(5), 1492.
[2] Abou-Chacra, R., Thouless, D. J., & Anderson, P. W. (1973). A selfconsistent theory of localization. Journal of Physics C: Solid State Physics, 6(10), 1734.
[3] Pietracaprina, F., Ros, V., & Scardicchio, A. (2016). Forward approximation as a mean-field approximation for the Anderson and many-body localization transitions. Physical Review B, 93(5), 054201.
[4] Basko, D. M., Aleiner, I. L., & Altshuler, B. L. (2006). Metal–insulator transition in a weakly interacting many-electron system with localized single-particle states. Annals of physics, 321(5), 1126-1205.
[5] Ros, V., Müller, M., & Scardicchio, A. (2015). Integrals of motion in the many-body localized phase. Nuclear Physics B, 891, 420-465.
[6] Panda, R. K., et al. (2020). Can we study the many-body localisation transition?. EPL (Europhysics Letters), 128(6), 67003.

Requested changes

1- Connect with the locator expansion
2- Clarify and/or extend the discussion in the subsection "Connections to the many-body localisation finite-size crossover"

  • validity: high
  • significance: high
  • originality: high
  • clarity: good
  • formatting: perfect
  • grammar: perfect

Author:  Philip Crowley  on 2022-01-04  [id 2067]

(in reply to Report 1 on 2021-07-08)

We thank the referee for their time, their positive report, and their recommendation to publish the manuscript with minor amendments.

We respond to the referee's comments inline, and include with this report, an amended manuscript with the major alterations highlighted.

"The paper studies the different coupling regimes of a single spin to a bath, the latter described by either one of the Gaussian ensembles or a random matrix with Poisson statistics or a realistic model given by an ergodic spin chain."

"The main object of study is the susceptibility $\chi$ in Eq. (3) or (28)."

"This is a random variable good to study the response of a system to the introduction of a local operator, which has been studied by several people before. In the specific case at hand, I think the authors fail to recognize that this is exactly the first term in a locator expansion in the spirit of [1] and [2] and more recently in several other papers (see the more recent [3]).
In [1-3] the denominators in the energy are uncorrelated, while in principle $E_a$, $E_b$ in this paper are, for some ensembles, correlated. But the long tail
$f \sim \chi^{-3/2}$ which is a main feature of the distribution, follows from small denominators. The small denominators come from pairs $a$, $b$ for which $E_a - E_b \sim h_s = O(1)$ but since the level spacing $\delta \ll 1 \sim h_s$ (which is the scale on which $E_a$, $E_b$ correlate), this means that $E_a$, and $E_b$ are uncorrelated to a very good approximation. So we fall back in the conditions of [1,2], and the tail follows from the discussion after eq. (5.1) in [2]. Analogous results are in the denominator analysis in [4,5]."

"I think the authors should recognize this fact, which does not subtract to their analysis (which goes beyond this point, significantly) but it connects with a set of papers which the authors have not involved in their discussion. I was in particular surprised of the absence of reference to [4] in their otherwise very generous bibliography, considering this work has significant implications for MBL, for which the authors have a separate sub-section."

We thank the referee for raising this connection. The referee is correct about the close connection between our analyses to the locator expansion, and the relation of our results to those previously obtained in this literature. We are indeed clear in Sec 3.1 that we are performing an expansion in the coupling between two otherwise decoupled subsystems. However, in order to further make the connection explicit in the revised manuscript we have included a discussion of this point at the end of Sec 3.1, and after point (i) in Sec 3.2.2, where we emphasise that the $f \sim \chi^{-3/2}$ result can be understood in this context of the locator expansion. In each of these places we have included the references suggested by the referee.

"Regarding the issue of whether ETH should be modified or not in the thermalizing region preceding the MBL transition is not clear to me if the authors agree or not with their references (in the paper [32,53]) and if they agree or disagree with [6] (in this report) which seems to find that if one goes to sufficiently large system sizes, the distribution of the off-diagonal matrix elements return gaussian and that this is not in contradiction with having subdiffusive transport."

The referee has asked for clarification regarding whether our results are in contradiction or agreement with Refs. [32,58] (reference numbers updated to correspond to most recent draft).

We first summarise the relevant findings of Refs. [32,58]. These studies focus on two distinct features of the off-diagonal matrix elements on the thermal side of the MBL transition in random Heisenberg type models, both of which the authors argue are consequences of a putative sub-diffusive thermal phase. The first observation concerns changes to the $L$ dependence of the spectral function, and the second regards deviations from Gaussianity in the distribution of off-diagonal matrix elements. We focus on the latter point, which we contest, and provide a different explanation of the observed non-Gaussianity.

Our analyses does not provide a complete model of an interacting many body system. Our results do nevertheless allow us to understand on the origin of the non-Gaussian distributions of off diagonal matrix elements in numerically accesible spin chains. Non Gaussianity results from the presence of spins which are insufficiently strongly coupled to their environment---irrespective of the level statistics of their environment. This explanation does not make reference to the presence, or otherwise, of sub-diffusion. Specifically, in sufficiently small systems, where spins are found in the intermediate regime ($J \sqrt{\chi_\star} \lesssim 1$), we predict off diagonal element distributions of the form of Fig. 11. This non-Gaussianity goes hand-in-hand with other features of the intermediate coupling regime, such as the spins having eigenstate entanglement entropies which are bi-modal, being either close to the thermal ($S= \log 2$) or localised ($S=0$) value. The presence of bi-modal spin entropies is inconsistent with a thermal phase, sub-diffusive or otherwise, in contrary to the second claim of Refs. [32,58]. Such bi-modal distributions have been observed numerically [66] at the same system sizes as Refs. [32,58]. Finally, Ref.[6](of the referees report)---pointed out by the referee---shows these distributions flow towards Gaussianity upon increasing the system size. The Spin-ETH model explains this flow via an increasing value of $J \sqrt{\chi_\star}$ felt by each of the spins in the chain.

Ref. [6] pointed out by the referee indicates, in agreement with our claim, that even in the putative sub-diffusive phase, the distribution of off-diagonal elements becomes Gaussian at sufficiently large system size.

In response to the referees comments we have expanded and clarified this point in the discussion section of the revised manuscript to avoid any confusion for the reader.

"For the rest of the paper I have no objections or comments. I like the treatment of the off-diagonal elements of the operators for the intermediate values of couplings, I think it is an original and nice addition to the ETH vulgata and it deserves to be published. The paper as a whole deserves publication, once the authors fix the discussion above."

We thank the referee for their positive recommendation.

"I caught only one typo: Fig. 12. Upper and lower panel should be left and right panel in the caption."

We thank the referee's keen eye, and have corrected this typo, amongst others.

---

## Round 2 · Referee Report · Anonymous (Referee 1) · 2022-1-6

Report

The authors have improved the text and have followed all the suggestions given in the previous report. Therefore I think the paper can be published as it is.

---

## Round 2 · Referee Report · Marco Rossi (Referee 2) · 2022-1-25

Report

The paper concerns a detailed study of the interaction of a two level system with a bath. The discussion is very general: various coupling regimes and different types of baths are studied, as the first referee remarks. In particular the main novelty is the discussion of the weak and intermediate coupling regimes, with the development of an ETH-like ansatz.

The paper is quite technical, but I appreciate that the results are proved and discussed with a remarkable accuracy. Non trivial checks of the statements are provided by use of numerical techniques and are illustrated in several useful pictures.

Therefore, in my opinion the paper deserves publication in the present form.

I just noticed some misprints:

1) at the beginning of page 6 should the open boundary condition involve \sigma ^x (and not \sigma ^z)? 2) five lines after formula (34): relates -->related 3) five lines before formula (53): The the --->The

---

## Round 2 · List of Changes

In the response to the referees comments we have:

  • Included discussion of the connection to the locator expansion (Sec 3.1 p.8, Sec 3.2.2 p.11)
  • Re-written discussion point "Connections to the many-body localisation finite-size crossover" to clarify the points discussed in our response to the referee's comments (Sec. 8 p.29).
  • Corrected typos.

---

## Editorial Decision

published